# LRRK2-mutant microglia and neuromelanin synergize to drive dopaminergic neurodegeneration in an iPSC-based Parkinson's disease model
Lucas Blasco-Agell[1,2,13], Meritxell Pons-Espinal [1,2,13], Veronica Testa [1,2,13], Gerard Roch [3,4,13], Jara Montero-Muñoz [1,2], Irene Fernandez-Carasa [1,2], Valentina Baruffi[1,2], Marta Gonzalez-Sepulveda [3,4], Yvonne Richaud-Patin[5,6], Senda Jimenez[5,6], Thais Cuadros [3,4], Joana M. Cladera-Sastre[3,4], Joan Compte [3,4], Zoe Manglano-Artuñedo[7], Salvador Ventura [7,8], Manel Juan [9], Eduardo Tolosa[10], Angel Raya [5,6,11]✉, Miquel Vila [3,4,11,12]✉ & Antonella Consiglio [1,2]✉

Parkinson's disease (PD) is a progressive, incurable neurodegenerative disorder characterized by the loss of neuromelanin (NM)-containing dopamine neurons (DAn) in the *substantia nigra* of the midbrain. Non-neuronal cells are increasingly recognized as contributors to PD. We generated human microglia-like cells (hMG) from induced pluripotent stem cells (iPSC) derived from patients with LRRK2 PD-causing mutations, gene-corrected isogenic controls, and healthy donors. While neither genotype induced neurodegeneration in healthy DAn, LRRK2 hMG become hyperreactive to LPS stimulation, exhibiting increased cytokine expression, reactive oxygen species, and phagocytosis. When exposed to NM-containing particles, but not α-synuclein fibrils, LRRK2 hMG trigger DAn degeneration, in a process that is prevented by pre-treatment with the immunomodulatory drug ivermectin. Finally, post-mortem analysis of midbrain tissue of LRRK2-PD patients show increased microglia activation around NM-containing neurons, confirming our in vitro findings. Overall, our work highlights NM-activated microglia's role in PD progression, and provides a model for testing therapeutic targets.

Parkinson's disease (PD) is an age-associated movement disorder resulting from progressive loss of neuromelanin (NM)-containing dopaminergic neurons (DAn) in the substantia nigra pars compacta (SNpc) and deposition of Lewy bodies (LBs), which mainly consist of α-synuclein[1]. In addition to these hallmark features, glial-mediated neuroinflammatory responses have emerged as key contributors to PD pathophysiology[2]. While most PD cases are idiopathic, meaning they have no known cause[3], pathogenic variants in PD-associated genes are identified in about 15% of patients[4], with *LRRK2* gene mutations representing the most common genetic cause of familial PD[5].

LRRK2 is a large, ubiquitously expressed protein, containing both GTPase and kinase domains, involved in autophagy-lysosomal pathway, modulation of cytoskeletal dynamics, vesicle trafficking, mitochondrial function and immunity[5]. Notably, LRRK2 expression has been found increased in several peripheral immune cells of PD patients compared to

healthy controls[6–9]. Additionally, its expression is upregulated in microglia following stimulation with lipopolysaccharides (LPS), IFN-γ, HIV-Tat protein, manganese, or rotenone[10–13].

Multiple lines of evidence suggest a correlation between mutant LRRK2 and several pathogenic mechanisms linked to PD initiation and progression, including an imbalanced inflammatory signalling[14]. However, little is known about the correlation between LRRK2- mutant microglia activation and PD-related endogenous stimuli, such as NM.

NM is a byproduct of dopamine metabolism that progressively accumulates with age, primarily in the SN DAn[15,16]. While the functional significance of NM production, if any, remains unclear, it has been hypothesized that NM may play a neuroprotective role by removing excessive cytosolic catecholamine oxidized species, chelating potentially toxic metals or sequestering environmental neurotoxins[17]. Despite its potential protective function, recent studies using humanized NM-

producing animal models have shown that age-related intracellular NM accumulation beyond a pathogenic threshold compromises neuronal function and viability, ultimately triggering PD-like pathology in both rodents and non-human primates (NHPs)[18–20]. Moreover, the release of NM from dying DAn in PD brains and NM-producing rodents/NHPs has been shown to initiate an early immune response in the brain, leading to inflammation and oxidative stress[21–25], both of which may contribute to PD onset and progression.

While studies suggest that NM-induced microglia activation can contribute to DAn cell death in primary microglia cultures and rat models[21,26], the impact of LRRK2 mutations on NM-driven microglia activation remains unclear. In addition, it is uncertain whether human-derived microglia respond to NM similarly to primary microglia.

To directly investigate the crosstalk between extracellular NM and microglial activation in the context of pathogenic LRRK2 mutations, we generated iPSC-derived microglia (hMG) from two PD patients carrying the LRRK2 G2019S mutation (L2-PD) along with their respective isogenic controls (L2-PD$^{corr}$) and healthy individuals (CTL). We consistently generated a population of ventral midbrain (vm) DAn from CTL-iPSC and characterized them for the expression of postmitotic dopaminergic markers. Subsequently, we established neuron/microglia co-cultures containing healthy iPSC-derived vmDAn together with hMG from all conditions. Exposing this system to purified NM extracts enabled us to investigate the potential role of microglia-mediated NM-inflammatory signalling in PD-related neuropathology. Using this experimental setup, we found that extracellular NM reduced the number of vmDAn in the presence of L2-PD-derived microglia and that this effect was reversed by immunomodulation with FDA-approved drug ivermectin (IVM). Moreover, the induced neurodegeneration was specific to NM, as the addition of α-synuclein pre-formed fibrils (PFF) to L2-PD hMG and vmDAn co-cultures did not produce the same effect.

Overall, our findings highlight the deleterious role of NM-activated L2-PD microglia and establish a human-relevant preclinical platform for immunomodulatory drug screening in the context of Parkinson's disease.

## Results

### iPSC-derived hMG express typical microglial markers

We generated human microglia cells (hMG) from iPSC lines following a serum- and feeder-free four-step differentiation protocol[22,23] (Supplementary Fig. 1a). The iPSC lines were derived from two LRRK2-PD patients carrying the G2019S mutation (L2-PD), one healthy donor (CTL), and two isogenic iPSC lines in which the LRRK2 G2019S missense mutation was corrected (L2-PD$^{corr}$) (Table 1).

Immunocytochemistry (ICC) detection of key microglia markers confirmed robust expression of IBA-1, CX3CR1, and TMEM119 in all iPSC-derived hMG after one week in culture (Fig. 1a, b and Supplementary Fig. 1b). No contamination from other cell types, such as astrocytes or neurons, was observed, as assessed by immunostaining with anti-GFAP or TUJ1 antibody, respectively (Fig. 1a and Supplementary Fig. 1b). Additionally, hMG expressed the myeloid marker CD11b as well as the scavenger receptor CD163 (Supplementary Fig. 1c, d), further confirming their microglial identity. Moreover quantitative RT-PCR of canonical human microglia-specific genes, including *P2RY12*, *GAS6*, *PROS1*, *MERTK*, *C1Qa*,

and *GPR34*, confirmed their expression in hMG, but not in iPSC[24] (Supplementary Fig. 1e). Importantly, no differences or evidence of a mutation effect were observed in hMG generation capacity between lines. In addition, when included into ventral midbrain organoids (vmO) to simulate a more physiological environment, hMG adopted a more complex morphology, characterized by increased ramifications (Supplementary Fig. 2a–c).

Collectively, these results confirm the successful generation of microglia-like cells, providing a continuous source of human CTL, L2-PD$^{corr}$ and L2-PD microglia for subsequent analyses.

### Functional validation of hMG

Next, we aimed at validating hMG using functional and physiological assays. First, we assessed the secretion of IL-1β, IFNγ, IL-6 and TNFα, pro-inflammatory cytokines known to be elevated in PD patients[25,27–31], by stimulating hMG with Lipopolysaccharide (LPS), a well-established activator of microglial inflammation (Fig. 1c).

As expected, all iPSC-derived hMG secreted high levels of these pro-inflammatory cytokines upon LPS stimulation (Fig. 1d–f and Supplementary Fig. 2d). However, L2-PD hMG released significantly higher levels of IL-1β, IL-6, and TNFα compared to CTL and L2-PD$^{corr}$ hMG (Fig. 1d–f).

Additionally, L2-PD hMG exhibited increased expression of inflammasome-related genes compared to CTL and L2-PD$^{corr}$ hMG (Supplementary Fig. 2e, f), suggesting an enhanced inflammatory response associated with the LRRK2 G2019S mutation.

Given the role of microglia in synaptic pruning[32–34], we assessed the ability of hMG to phagocytize human synaptosomes (SYNs) isolated from iPSC-derived mature neurons (Supplementary Fig. 2g–j). Quantification of pHrodo-fluorescent labelled SYNs confirmed efficient engulfment of SYNs by all iPSC-derived hMG lines. However, L2-PD hMG exhibited significantly increased SYN phagocytosis compared to CTL and L2-PD$^{corr}$ hMG (Fig. 1g, h), indicating that the enhanced phagocytic activity is linked to the LRRK2 G2019S mutation.

### L2-PD hMG increases NM uptake and releases ROS

PD neuropathology is characterized by the preferential degeneration of neuromelanin (NM)-containing dopaminergic neurons (DAn)[26,35] and increased microgliosis[36–38]. To define the interplay between microglia and extracellular NM released from dying neurons (rNM) in PD-related neurodegeneration, we first examined the effects of purified NM on microglial activation. To do that, hMG were exposed to NM extracts purified from a neuronally differentiated SH-SY5Y cell line, which produces human-like NM through overexpression of the melanin-producing enzyme human tyrosinase[18,39] (Supplementary Fig. 3a). Microglial responses were then assessed using time-lapse imaging, qPCR, and ROS analyses (Fig. 2a). Live imaging revealed that hMG were highly motile, actively scanning their microenvironment and extending filopodia to interact with and phagocytose NM particles (Supplementary Movie 1). Sixteen hours after NM exposure, hMG accumulated NM, with L2-PD hMG displaying a significantly higher uptake of NM particles compared to CTL and L2-PD$^{corr}$ hMG (Fig. 2b, c and Supplementary Fig. 3b). Additionally, L2-PD hMG exhibited increased motility, covering greater distances (Fig. 2d, e and Supplementary Fig. 3c) than their corresponding controls.

**Table 1 | Summary of the healthy controls and patients used in this study**

| Cell line code | Status | Code used in this study | Sex | Age at donation | Age at onset | LRRK2 mutation | Isogenic control |
|---|---|---|---|---|---|---|---|
| SP-11 | Control | CTL | F | 48 | – | No | – |
| SP-09 | Control | CTL | M | 66 | – | No | – |
| SP-12 | L2-PD | L2-PD1 | F | 63 | 49 | G2019S | – |
| SP-13 | L2-PD | L2-PD2 | F | 68 | 57 | G2019S | – |
| SP-12 wt/wt | Isogenic Control | L2-PD1$^{corr}$ | F | 63 | – | G2019S | LRRK2$^{G2019S}$ corrected |
| SP-13 wt/wt | Isogenic Control | L2-PD2$^{corr}$ | F | 68 | – | G2019S | LRRK2$^{G2019S}$ corrected |

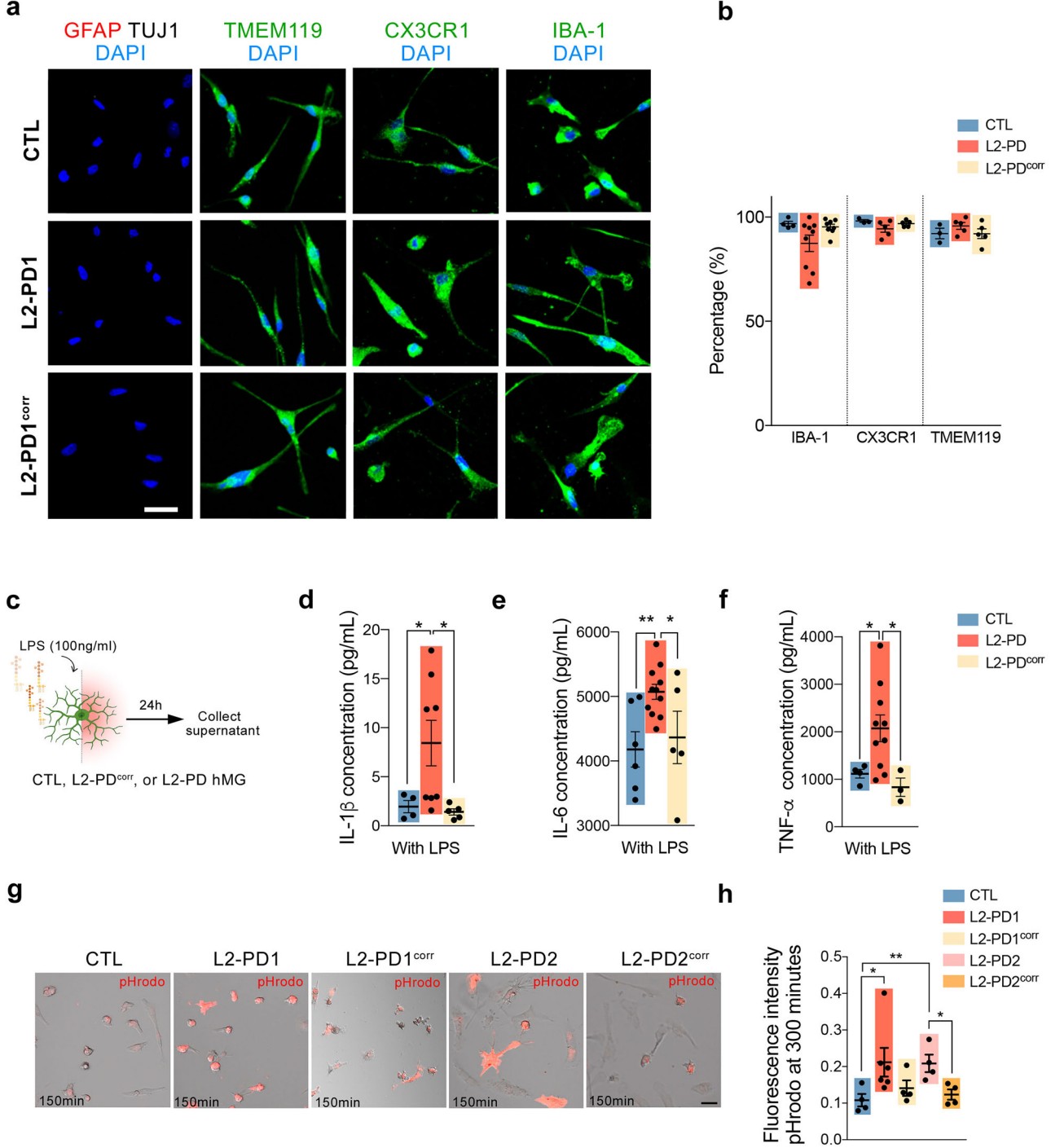

**Fig. 1 | Characterization of hMG cells. a** Representative ICC images of iPSC-derived hMG after 7 days in culture from CTL (SP09), L2-PD (L2-PD1: SP12) and L2-PD^corr^ (L2-PD1^corr^: SP12wt/wt) iPSC lines staining positive for IBA-1, CX3CR1 or TMEM-119 (green) and negative for astrocytic (GFAP) or neuronal (TUJ1) markers. Nuclei are counterstained with DAPI (blue). Scale bar = 30 μm. **b** Percentage of hMG cells from CTL (SP09), L2-PD (L2-PD1: SP12; L2-PD2: SP13) and L2-PD^corr^ (L2-PD2^corr^: SP13wt/wt) lines expressing IBA-1, CX3CR1 or TMEM-119 in respect to DAPI. Individual data plotted, along with mean ± SEM. N = 3 of independent experiments, each experiment containing two technical duplicates. **c** Schematic representation of LPS stimulation (100 ng/ml). Cytokine profile for IL-1β (**d**), IL-6 (**e**) and TNF-α (**f**) of hMG from CTL (SP09), L2-PD (L2-PD1: SP12; L2-PD2: SP13) and L2-PD^corr^ (L2-PD2^corr^: SP13wt/wt) iPSC lines after 24 hours of LPS stimulation.

Individual data plotted, along with mean ± SEM. One-way ANOVA with Uncorrected Fisher LSD test. N = 3 of independent experiments, each experiment containing two technical duplicates. **g** Representative Bright Field and Red fluorescent images of CTL (SP09), L2-PD1 (SP12), L2-PD1^corr^ (SP12wt/wt), L2-PD2 (SP13), and L2-PD2^corr^ (SP13wt/wt) hMG phagocyting pHrodo labelled SYNs after 150 minutes of exposure to SYNs (Scale bar = 25 μm). **h** pHRodo fluorescence intensity within hMG after 300 minutes comparing CTL (SP09), L2-PD1 (SP12), L2-PD1^corr^ (SP12wt/wt), L2-PD2 (SP13), and L2-PD2^corr^ (SP13wt/wt) hMG. Individual data plotted, along with mean ± SEM. Kruskal-Wallis non-parametric test with Uncorrected Dunn's test. N = 3 of independent experiments, for some experiments two technical duplicates were measured. *$p < 0.05$, **$p < 0.01$. p-values over 0.1 (non-significant) are not shown.

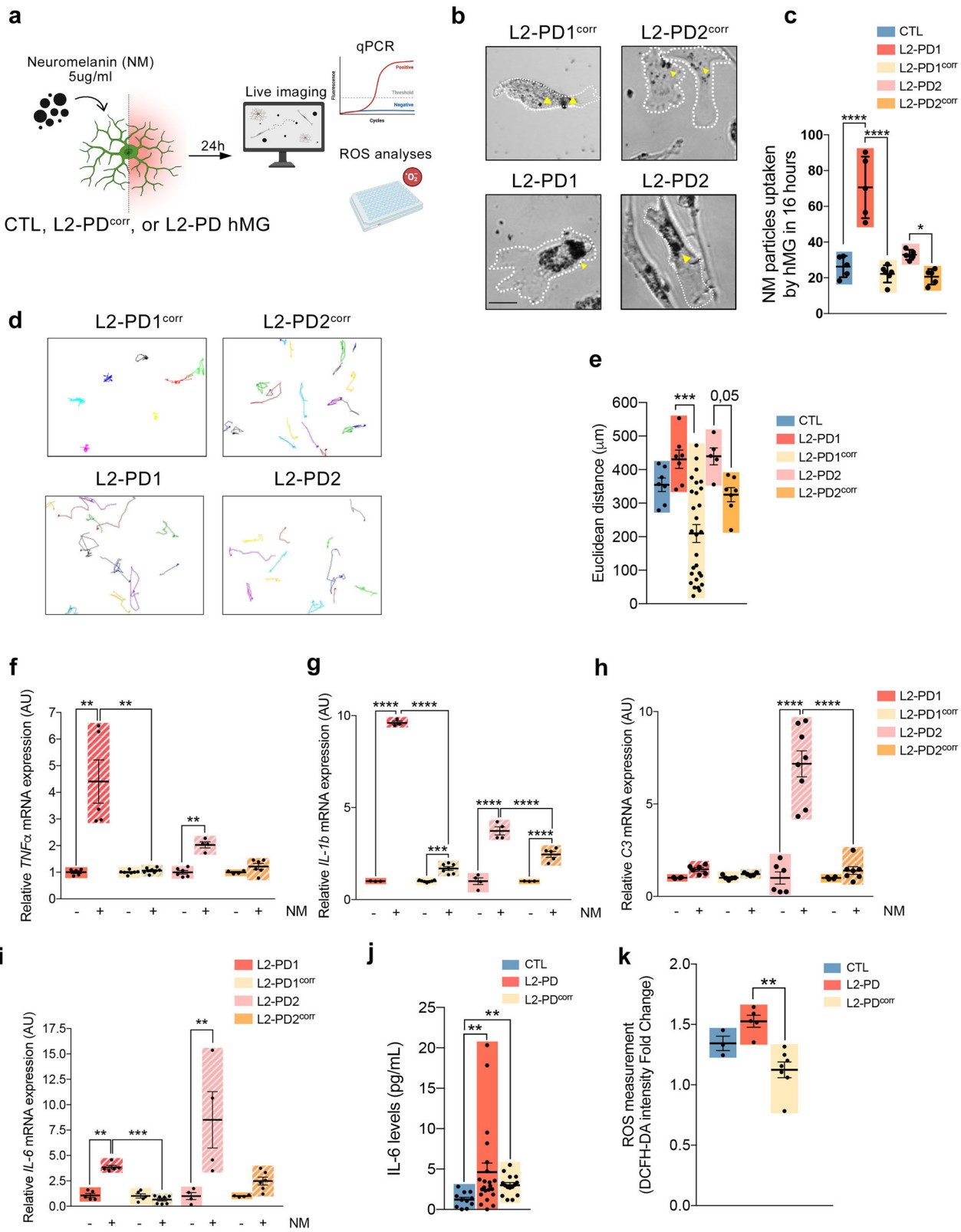

To determine whether hMG can be activated by NM, cells were cultured with and without NM for 24 h, after which pro-inflammatory and neurotoxic gene expression levels were assessed. We found that L2-PD hMG exhibited an increase in *IL-1β, IFNγ, IL-6, C3* and *TNFα* mRNA expression as compared to their isogenic controls upon NM exposure (Fig. 2f–i and Supplementary Fig. 3d). Consistent with these findings, NM stimulation led to a significantly increased release of IL-6 from L2-PD hMG compared to CTL hMG, with levels reduced in their isogenic counterpart (Fig. 2j). To further assess hMG activation following NM exposure, we measured dichlorofluorescein (DCF) fluorescence as an indicator of reactive oxygen species (ROS) production. We found that NM increased ROS production in L2-PD hMG as compared to control and L2-PD^corr hMG (Fig. 2k).

**Fig. 2 | L2-PD hMG increase neuromelanin particle phagocytosis and induce the expression of pro-inflammatory cytokines and ROS. a** Schematic representation of NM stimulation (5 ug/ml) and following analyses. **b** Representative Bright Field images of L2-PD1 (SP12), L2-PD1$^{corr}$ (SP12wt/wt), L2-PD2 (SP13), and L2-PD2$^{corr}$ (SP13wt/wt) hMG phagocyting NM particles for 16 hours (yellow arrow-heads for phagocyted particles; Scale bar = 25 μm). **c** NM particles uptaken by hMG over 16 hours comparing CTL (SP09), L2-PD1 (SP12), L2-PD1$^{corr}$ (SP12wt/wt), L2-PD2 (SP13), and L2-PD2$^{corr}$ (SP13wt/wt) hMG. Individual data plotted, along with mean ± SEM. One-way ANOVA with Uncorrected Fisher LSD test. N = 3 of independent experiments, for some experiments two technical duplicates were measured. **d** Spontaneous migration paths of L2-PD1 (SP12), L2-PD1$^{corr}$ (SP12wt/wt), L2-PD2 (SP13), and L2-PD2$^{corr}$ (SP13wt/wt) hMG. hMG were tracked for 16 h. The location of each cell was determined every 2 minutes and connected to depict its migration route. **e** Quantification of Euclidean distances (μm) comparing CTL (SP09), L2-PD1 (SP12), L2-PD1$^{corr}$ (SP12wt/wt), L2-PD2 (SP13), and L2-PD2$^{corr}$ (SP13wt/wt) hMG. Individual data plotted, along with mean ± SEM. Kruskal-Wallis non-parametric test with Uncorrected Dunn's test. N = 3 of independent experiments, each dot represents a cell. Relative mRNA expression of pro-inflammatory cytokines TNF-α (**f**), IL-1β (**g**), C3 (**h**), and IL-6 (**i**) in L2-PD1 (SP12), L2-PD1$^{corr}$

(SP12wt/wt), L2-PD2 (SP13), and L2-PD2$^{corr}$ (SP13wt/wt) hMG at basal conditions or under NM stimulation for 24 hours. Individual data plotted, along with mean ± SEM. One-way ANOVA with Tukey multiple comparison test for (**g**) and (**h**); Kruskal-Wallis non-parametric test with Uncorrected Dunn's test for (**f**) and (**i**). N = 3 of independent experiments, each experiment containing two technical duplicates. **j.** Levels of IL-6 released by hMG from CTL (SP09), L2-PD (L2-PD1: SP12; L2-PD2: SP13), and L2-PD$^{corr}$ (L2-PD1$^{corr}$: SP12wt/wt; L2-PD2$^{corr}$ SP13wt/wt) after being exposed to NM for 24 h. Individual data plotted, along with mean ± SEM. Kruskal-Wallis non-parametric test with Uncorrected Dunn's test. N = 3 of independent experiments, each experiment containing two technical duplicates. **k** Intracellular ROS by DCFH-DA intensity after NM stimulation in CTL (SP09), L2-PD (L2-PD1: SP12; L2-PD2: SP13), and L2-PD$^{corr}$ (L2-PD1$^{corr}$: SP12wt/wt; L2-PD2$^{corr}$ SP13wt/wt) hMG. Results are represented as a fold change between NM stimulation for 24 hours over non-stimulated conditions. Individual data plotted, along with mean ± SEM. One-way ANOVA with Tukey's multiple comparison test. N = 3 for CTL, N = 5 for L2-PD, and N = 7 for L2-PD$^{corr}$ of independent experiments. *p < 0.05, **p < 0.01, ***p < 0.001, ****p < 0.0001. p-value is specified for values between 0.05 and 0.1. p-values over 0.1 (non-significant) are not shown.

Taken together, these findings indicate that L2-PD hMG become activated upon NM engulfment, leading to increased expression of pro-inflammatory cytokines and elevated ROS release. This opens the possibility of exploring their detrimental effects on DAn survival.

## L2-PD hMG induces dopaminergic neuronal degeneration upon NM exposure

Next, to investigate the interaction between DAn and hMG and assess the impact of purified NM on DAn survival, we established a 2D neuron/microglia co-culture system. VmDAn were differentiated from a CTL iPSC line following a previously published protocol[40] (Supplementary Fig. 4a). By day 12 of differentiation, neural progenitor cells were committed towards a dopaminergic fate as shown by the expression of LMX1A and FOXA2 (Supplementary Fig. 4b). After 35 days of differentiation, vmDAn were considered mature, as evidenced by their expression of the dopaminergic marker TH and the mature neuronal marker MAP2 (Supplementary Fig. 4c). Moreover, vmDAn were positive for TH and FOXA2, confirming their dopaminergic identity (Supplementary Fig. 4d). The differentiation efficiency was approximately 47%, based on the proportion of TH+ neurons relative to the total neuronal population (MAP2 + ), while vmDAn accounted for 23% of the total cell population (Supplementary Fig. 4e, f).

After generating vmDAn, we cultured them on top of a human astrocyte feeder layer for 7 days. In a subset of experiments, we used a LV expressing GFP to visualize transduced microglia (Fig. 3a and Supplementary Fig. 5a, b).

In the absence of NM, IBA1-positive hMG displayed a highly ramified morphology and were observed in close proximity to TH-stained vmDAn (Supplementary Fig. 5c–e). Notably, vmDAn co-cultured with L2-PD or L2-PD$^{corr}$ hMG remained well-ramified and healthy, with no alterations in DAn survival (Fig. 3b, c). These findings suggest that L2-PD hMG do not negatively impact TH+ neurons under homeostatic conditions.

We next assessed the impact of NM-exposed hMG on the survival of CTL iPSC-derived vmDAn. After eight hours of NM treatment (Fig. 3a), both L2-PD and L2-PD$^{corr}$ hMG exhibited a shift toward a rounded, amoeboid morphology, characteristic of microglial activation (Supplementary Fig. 5c–e). Remarkably, only NM-activated L2-PD hMG significantly reduced the percentage of TH+ DAn, compared to untreated conditions (Fig. 3d–f, h and Supplementary Fig. 4g–j). Additionally, surviving vmDAn exhibited reduced branching, indicative of morphological impairment (Supplementary Figs. 4g–i and 5f, g). Similar neurodegenerative effects were observed when iPSC-derived vmDAn were treated with NM-exposed L2-PD hMG conditioned medium (MCM) (Supplementary Fig. 5h–j). These findings highlight a pathogenic role of NM-activated L2-PD hMG in DAn degeneration.

In order to determine whether hMG-induced neuronal degeneration was specific of NM activation, we treated vmDAn with preformed α-synuclein PFF for eight hours -the timeframe required to observe DAn degeneration in NM-treated co-cultures-, either alone or in the presence of L2-PD hMG (Supplementary Fig. 6a). While NM-treated co-cultures led to TH+ neuronal degeneration within eight hours, PFF treatment- even in the presence of L2-PD hMG - did not induce vmDAn degeneration (Supplementary Fig. 6b, c).

Altogether, these results suggest that L2-PD hMG drives DAn neurodegeneration in response to PD-associated stimuli like NM, but not α-synuclein PFF.

## Treatment with Ivermectin prevents L2-PD hMG induced dopaminergic neuronal degeneration upon NM exposure

Next, we investigated whether ivermectin (IVM), an immunomodulatory drug known to switch microglia into an anti-inflammatory phenotype[41,42], could attenuate NM-induced vmDAn degeneration in our co-culture system. VmDAn derived from CTL iPSC were co-cultured with L2-PD hMG, exposed to NM and treated with 3 μM IVM for 6 hours (Fig. 3a). We found that, unlike untreated conditions, IVM significantly prevent vmDAn degeneration induced by NM-activated L2-PD hMG (Fig. 3g, h). Consistent with a reduction in L2-PD hMG activation following IVM treatment, we observed a significant reduction in the expression of the pro-inflammatory cytokines *IL-1β, IL-6, TNFα*, and *C3* in L2-PD hMG monocultures (Fig. 3i–l).

Overall, our findings support the concept that microglia activated by extracellular NM may directly contribute to PD neurodegeneration and that immunomodulatory interventions in the early stages of PD could provide therapeutic benefits.

## Microglial activation is increased in post-mortem brains from L2-PD patients

To determine the relevance of our in vitro findings using iPSC-derived hMG to human PD pathology, we next examined microglial activation in the SN of post-mortem brains from iPD and L2-PD patients, comparing them to age-matched controls (Table 2). The number of IBA1+ cells was found to be significantly higher in both iPD and L2-PD brains compared to control subjects (Fig. 4a, b).

More importantly, a significantly greater proportion of these microglial cells exhibited an ameboid morphology—indicative of activation—in L2-PD cases compared to non-PD controls (Fig. 4c). Interestingly, the number of ameboid cells in iPD patients was intermediate between that of the control and L2-PD groups. In contrast, the number of non-reactive (ramified) microglial cells did not differ among the groups (Fig. 4d).

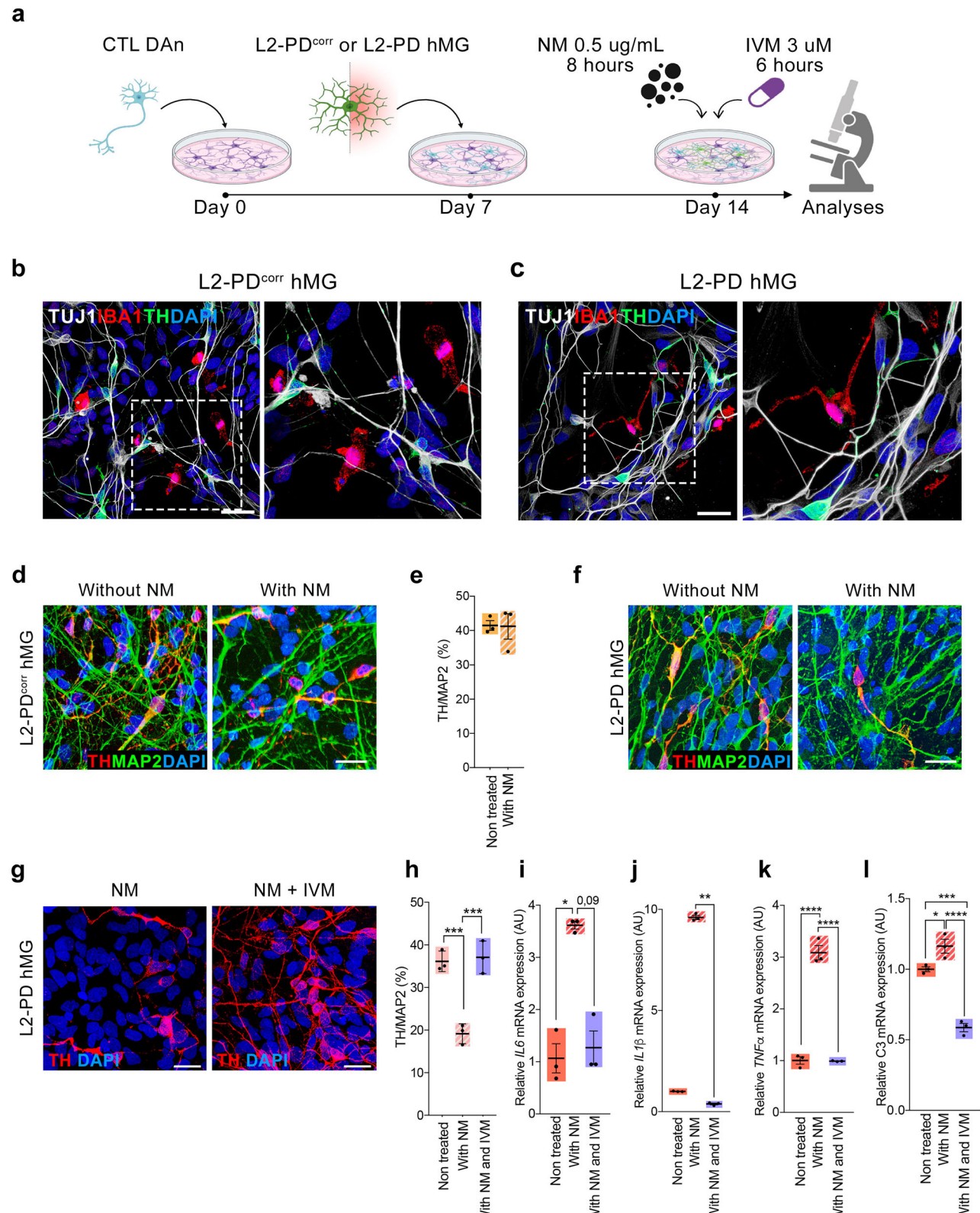

We then sought to determine whether rNM proximity influences microglial activation differently across groups. To that end, areas deemed distant from and in close proximity to eNM were analyzed and compared (Fig. 4f). As expected, ameboid IBA1+ cell density showed an increasing trend around rNM in all three groups. However, this effect was significantly more pronounced in L2-PD compared to both iPD and control groups (Fig. 4e). These results support our in vitro findings in NM-exposed hMG

and further reinforce the idea that NM-linked microglial activation may contribute to PD pathogenesis.

## Discussion

Neglected for decades, inflammation has gained attention in the latest years as a potential modulatory approach for PD[2,43–45]. As the primary immunocompetent cells of the CNS, microglia have become a key focus in

**Fig. 3 | L2-PD hMG exposed to NM induce dopaminergic degeneration in an iPSC-derived 2D neuron/hMG co-culture system. a** Schematic representation of the co-culture system and neuromelanin (NM) or Ivermectin (IVM) stimulation. Briefly, CTL (SP11) vmDAn were plated on top of astrocytes feeder layer on Day 0. One week later, L2-PD (L2-PD2: SP13) or L2-PD$^{corr}$ (L2-PD2$^{corr}$: SP13wt/wt) hMG were plated on top of vmDAn and exposed to NM at Day 14 for 8 hours, with or without treatment with IVM for 6 hours. Representative ICC images of L2-PD$^{corr}$ (L2-PD2$^{corr}$: SP13wt/wt) (**b**) or L2-PD2 (L2-PD2: SP13) (**c**) IBA1+ hMG (Red) in culture with TH + CTL vmDAn (SP11, Green) under basal conditions. In white TUJ1 + neurons (scale bar = 25 μm). **d** Representative images of CTL TH+ vmDAn (SP11) upon culture with L2-PD$^{corr}$ hMG (L2-PD2$^{corr}$: SP13wt/wt), without or with NM (scale bar = 30 μm). **e** Quantification of percentage of TH+ population over the total number of MAP2+ neurons in culture with L2-PD$^{corr}$ hMG (L2-PD2$^{corr}$: SP13wt/wt), without or with NM. Individual data plotted, along with mean ± SEM. Mann-Whitney test. N = 3 independent experiments. **f** Representative images of CTL TH+ vmDAn (SP11) upon culture with L2-PD hMG (L2-PD2: SP13), without or with NM (scale bar = 30 μm). **g** Representative images of CTL vmDAn (TH +, SP11) cultured with L2-PD hMG (L2-PD2: SP13) upon addition of NM or treatment with IVM and NM (Scale bar=25 μm). **h** Quantification of percentage of TH+ population over the total number of MAP2+ neurons in culture with L2-PD hMG (L2-PD2: SP13), under basal condition, upon addition of NM or treatment with IVM and NM. Individual data plotted, along with mean ± SEM. One-way ANOVA with Tukey multiple comparison test. N = 3 independent experiments. Relative mRNA expression of pro-inflammatory cytokines IL-6 (**i**), IL-1β (**j**), TNFα (**k**) and C3 (**l**) in L2-PD (L2-PD1: SP12) hMG at basal level, upon addition of NM, or upon treatment with IVM and NM. Individual data plotted, along with mean ± SEM. N = 3 independent experiments. One-way ANOVA with Tukey multiple comparison test for (**k**) and (**l**); Kruskal-Wallis non-parametric test with Uncorrected Dunn's test for (**i**) and (**j**). *p < 0.05, **p < 0.01, ***p < 0.001. ****p < 0.0001. p-value is specified for values between 0.05 and 0.1. p-values over 0.1 (non-significant) are not shown.

understanding how inflammation influences PD pathophysiology. Among various stressors, NM has been shown to induce microglial activation and trigger neuronal cell death[18-21,26,35,46-49]. However, due to methodological limitations and the limited availability of human samples, the impact of NM on microglial activation in PD has been only partially explored using non-human models[18-21,26]. The advent of iPSC technology has helped overcome these limitations, providing an alternative to traditional methods and enabling the generation of human-derived microglia-like cells through diverse protocols. A major advantage of iPSCs lies in their ability to capture the genetic diversity of individuals, offering insights into patient-specific disease phenotypes. However, line variability can represent a challenge to isolate the effects of specific mutations[50]. To address this, the use of isogenic lines, where only the mutation of interest is altered and the genetic background is identical, has become an essential control. This approach minimise confounding factors due to inter-individual genetic variability, strengthening the conclusions drawn from disease modelling studies[51].

Using an iPSC-based model, in this study we investigated the interaction between LRRK2-mutated microglia and dopamine neurons in the presence of NM. We first demonstrated that all iPSC-derived hMG (from one control, two LRRK2-PD and two corrected isogenic-PD iPSC lines) expressed several key microglial markers, confirming their bona fide microglia identity[52]. Moreover, hMG generated from all iPSC exhibited phagocytic activity, high motility, activation, and cytokine release in response to various stimuli. Compared to their isogenic counterparts, displayed significantly higher motility and phagocytic capacity—an aspect previously debated[10,14,53-55]. Moreover, upon LPS stimulation[11,14,56,57], we confirmed that L2-PD hMG exhibited an exaggerated inflammatory response.

Although previous reports have shown that NM can induce microglial activation in post-mortem brains[58-61] and experimental animal models[18-21,26,35,46-49], our findings reveal that NM can be actively phagocyted by human microglia, triggering ROS production and cytokine release. Importantly, this effect was confirmed in microglia derived from two independent LRRK2-PD iPSC, highlighting that the proinflammatory phenotype is dependent on the *G2019S* mutation.

Using a vmDAn-MG co-culture system we demonstrate that NM, but not α-synuclein PFF, induces neurodegeneration in TH+ neurons only when L2-PD hMG is present. This is partially due to the fact that L2-PD hMG, upon NM stimulation, enhance the production of pro-inflammatory cytokines. This finding is also in line with previous in vivo data from a rodent model producing human-like NM, which suggested that progressive NM accumulation with age leads to an early inflammatory response associated with neuronal dysfunction and degeneration[18,35]. Consistently, treatment of the co-culture with the immunomodulatory drug ivermectin, which rescues the increased cytokine levels induced by NM in L2-PD hMG, prevented NM-induced neurodegeneration. Given that ivermectin has also a direct neuroprotective role[62,63], future studies will explore its potential to rescue neurotoxicity induced by NM and LRRK2 mutations in hMG cells, avoiding its direct contact with dopaminergic neurons.

Hence, while the full mechanisms behind the inflammatory cascade caused by NM-induced neuroinflammation remain to be elucidated, our findings suggest that modulating inflammation in the degenerating substantia nigra pars compacta could be a therapeutic approach for PD patients, with the potential to halt or delay disease progression.

Interestingly, results from post-mortem human brains show that the LRRK2-PD group exhibits increased microgliosis and microglial activation, further supporting our in vitro data. Notably, the NM-induced microglial reaction is more pronounced in LRRK2-PD brains than in the iPD group, highlighting the heterogeneous nature of inflammatory responses in the PD context.

While we provide direct evidence that NM and LRRK2-mutant microglia contribute to PD-associated neurodegeneration, our study does not examine whether dopamine neurons susceptibility changes under pathological conditions, as PD neurons were not included in our system. In previous studies, we demonstrated that neurons derived from patient iPSCs recapitulate PD-relevant disease-associated phenotypes, including morphological alterations such as a reduced number of neurites, decreased neurite arborization, and the accumulation of autophagic vacuoles—features not observed in dopaminergic neurons (DAn) differentiated from control iPSCs[64]. Based on these findings, here we focused specifically on MG-dependent mechanisms and demonstrated that NM and LRRK2-mutant microglia exert non-cell autonomous neuronal dysfunction on dopaminergic neurons.

Overall, our data suggest that L2-PD mutant microglia are not pathogenic per se, but can become deleterious when overstimulated, specifically by NM released from dying neurons. These findings may have important therapeutic implications for early-stage intervention, where small amounts of extracellular NM are sufficient to trigger a significant inflammatory response before overt neurodegeneration occurs[49]. Moreover, the established system could serve as a platform to test new therapeutical targets in PD.

## Experimental procedures

**iPSC derived microglia (hMG).** In order to generate hMG cells from iPSCs, we have followed an already published serum- and feeder-free four-step protocol described by Douvaras et al.[22] built on previous studies from Yanagimachi et al.[65] and adapted by Mesci et al.[23]. A summary of the iPSC used is described in Table 1. The generation and use of human iPSCs in this work were approved by the Spanish competent authorities (Commission on Guarantees concerning the Donation and Use of Human Tissues and Cells of the Carlos III National Institute of Health). All procedures adhered to internal and EU guidelines for research involving derivation of pluripotent cell line. Informed consent was obtained from all patients using forms approved by the Ethical Committee on the Use of Human Subjects in Research at Hospital Clinic in Barcelona". All QC tests performed on iPSC lines can be found in https://doi.org/10.5281/zenodo.16601152. Unless specified, all growth factors, small molecules and cytokines employed during this project were

**Article**

## Table 2 | Human sample information

| Case | Gender | Age at death (years) | Disease duration (years) | PMI (hr) | Braak PD staging |
|---|---|---|---|---|---|
| Control-1 | Female | 59 | N/A | 2 | N/A |
| Control-2 | Male | 59 | N/A | 6 | N/A |
| Control-3 | Male | 59 | N/A | 5 | N/A |
| Control-4 | Male | 60 | N/A | N/A | N/A |
| Control-5 | Female | 60 | N/A | N/A | N/A |
| Control-6 | Female | 61 | N/A | N/A | N/A |
| Control-7 | Female | 62 | N/A | 2 | N/A |
| Control-8 | Male | 63 | N/A | N/A | N/A |
| Control-9 | Male | 63 | N/A | 12 | N/A |
| Control-10 | Female | 65 | N/A | N/A | N/A |
| Control-11 | Male | 65 | N/A | N/A | N/A |
| Control-12 | Female | 66 | N/A | N/A | N/A |
| Control-13 | Male | 69 | N/A | N/A | N/A |
| Control-14 | Female | 70 | N/A | 14 | N/A |
| Control-15 | Female | 70 | N/A | N/A | N/A |
| Control-16 | Female | 71 | N/A | N/A | N/A |
| Control-17 | Female | 73 | N/A | N/A | N/A |
| Control-18 | Female | 74 | N/A | N/A | N/A |
| Control-19 | Male | 76 | N/A | 11.5 | N/A |
| Control-20 | Male | 78 | N/A | 4 | N/A |
| Control-21 | Female | 79 | N/A | N/A | N/A |
| Control-22 | Female | 79 | N/A | N/A | N/A |
| Control-23 | Male | 81 | N/A | 2 | N/A |
| Control-24 | Male | 83 | N/A | 13 | N/A |
| Control-25 | Female | 83 | N/A | 7.2 | N/A |
| Control-26 | Female | 83 | N/A | 4 | N/A |
| Control-27 | Male | 84 | N/A | 4 | N/A |
| Control-28 | Female | 86 | N/A | 4 | N/A |
| Control-29 | Female | 90 | N/A | 12.25 | N/A |
| Control-30 | Female | 91 | N/A | 8 | N/A |
| Control-31 | Male | 91 | N/A | N/A | N/A |
| Control-32 | Female | 94 | N/A | N/A | N/A |
| iPD-1 | Male | 76 | N/A | N/A | iPD |
| iPD-2 | Female | 77 | N/A | N/A | iPD |
| iPD-3 | Male | 77 | N/A | N/A | iPD |
| iPD-4 | Male | 77 | N/A | N/A | iPD |
| iPD-5 | Male | 80 | N/A | N/A | iPD |
| iPD-6 | Female | 81 | N/A | N/A | iPD |
| iPD-7 | Male | 81 | N/A | N/A | iPD |
| iPD-8 | Female | 85 | N/A | N/A | iPD |
| iPD-9 | Female | 88 | N/A | N/A | iPD |
| L2-PD-1 | Male | 84 | N/A | 20 | I-II |
| L2-PD-2 | Female | 78 | 26 | 27.5 | IV |
| L2-PD-3 | Female | 77 | 10 | 8 | N/A |
| L2-PD-4 | Female | 93 | 15 | 7 | IV |
| L2-PD-5 | Female | 69 | 17 | 12.5 | V |
| L2-PD-6 | Male | 69 | N/A | 15.5 | N/A |

*L2-PD* LRRK2-PD, *iPD* Idiopathic PD, *PMI* post-mortem interval (hours), *N/A* not available.

purchased from PeproTech®. Single iPSC colonies were manually passaged onto fresh Matrigel (Corning)-coated 6 cm$^2$ plastic plate (Thermo Fisher Scientific). After 2-4 days, protocol started by supplementing mTeSR™1 medium (Stem Cell Technology) with 80 ng/mL of Bone Morphogenetic Protein (BMP)-4 every day from day 0 to day 4. From day 5, SP34 medium was employed, which consisted of StemProTM-34 SFM (Gibco™) with 1% of P/S (Cultek) and 1% of Ultraglutamine (Glut; Lonza). For two days, SP34 medium was supplemented with 80 ng/ml of VEGF, 100 ng/ml of Stem Cell Factor (SCF) and 25 ng/ml of Fibroblast Growth Factor (FGF)-2. After day 7 until day 14, SP34 was supplemented with 50 ng/ml of Fms-like tyrosine kinase 3-Ligand (Flt3-L, Human-zyme, HZ-1151), 50 ng/ml of IL-3, 50 ng/ml of SCF, 5 ng/ml of Trom-bopoietin (TPO) and 50 ng/ml of M-CSF, changing the medium at day 10. The last step consisted on the addition to SP34 of Flt3-L, M-CSF and 25 ng/ml of GM-CSF from day 14, changing the medium every 3-4 days. At day 35, floating hMG progenitors were collected from the culture's supernatant and passed through a 70 μm Filcon™ Syringe-Type nylon mesh (BD Biosciences) to ensure a single-cell suspension and to remove cell clumps. Cells were counted, centrifuged at 300x*g* for 10 minutes and cultured in Roswell Park Memorial Institute (RPMI) 1640 Medium (Gibco™) with 1% of P/S and 1% of Glut, supplemented by M-CSF and 50 ng/ml IL-34. Medium was changed with fresh factors every 2-3 days. hMG progenitors were plated onto Matrigel-coated glass coverslips in 24-well plates (25,000 cells/well), on uncoated 96-well plates (10,000 cells/well) or 12-well plates (200,000 cells/well) (all from Thermo Fisher Scientific) depending on the experiment. hMG mature cells were cultured for one week before being used for experiments. For treatments, 100 ng/mL of LPS (Sigma-Aldrich) or 5 ug/mL of NM were added for a total of 24 hours.

**iPSC-derived vmDAn.** Generation of vmDAn from iPSCs was performed following the floor plate (FP) derivation protocol[66,67]. iPSC medium was changed to Serum Replacement Medium (SRM), which was based on KO-DMEM (Gibco™) with 15% of KO Serum Replacement (KO-SR; Gibco™), 1% of P/S, 1% of Glut, 1% of Non-Essential Amino Acids (NEAA; Lonza), and 10 μM of β-Mercaptoethanol (BME; Gibco™). The first day, SRM was supplemented with 100 nM of LDN193189 hydrochloride (LDN; Sigma-Aldrich) and 10 μM of SB 431542 (SB; Tocris Bioscience). From there and during all the differentiation, cells were maintained under hypoxic conditions (37 °C Temperature, 5% CO$_2$ and 5% O$_2$). From day 1 to day 5, SRM medium was supplemented with same concentrations of LDN and SB and adding 1 μM of Smoothened Agonist (SAG; Merck Millipore), 2 μM of Purmorphamine (Tocris Bioscience), and 100 ng/mL of FGF-8. From day 3 onwards, 3 μM of StemMACS™ CHIR99021 (CHIR; Miltenyi Biotec) were also added. From day 5, medium was progressively switched from SRM to N2, which consisted of Neurobasal™ Medium (Gibco™), 1x N2 supplement (Gibco™), 1% of P/S and 1% of Glut. On day 11, medium was changed with N2 supplemented with LDN and CHIR and cells could be either expanded as FP progenitors or matured to vmDAn. The additional step to expand, freeze and thaw FP progenitors was adapted from Fedele et al.[67]. Cells were maintained in N2 supplemented with LDN and CHIR at day 11, until reaching 90% of confluency within the plate. To differentiate FP progenitors into vmDAn, medium was changed to B27-vitA, which contained Neurobasal™ (Gibco™), 2x B27™- vitamin A (Gibco™), 1% of P/S and 1% of Glut. Every two days, medium was fully changed with B27 supplemented with 20 ng/mL of BDNF, 20 ng/mL of Glial cell-line Derived Neurotrophic Factor (GDNF), 1 ng/mL of TGF-β3, 10 μM of Deoxyadenosine Triphosphate (dAPT; Calbiochem), 0.5 mM of Dibu-tyryl cyclic-Adenosine Monophosphate sodium salt (cAMP; Sigma-Aldrich) and 0.2 mM of L-Ascorbic Acid (AA; Sigma-Aldrich). On day 20, cells were detached with Accutase (Merck Millipore), counted and centrifuged at 300x*g* for 5 minutes. Cells were plated on 6-well plates or glass coverslips in 24-well plastic plates with a triple-coating of 15 μg/mL Poly-L-Ornithine solution (Sigma-Aldrich), 3 μg/mL of Laminin

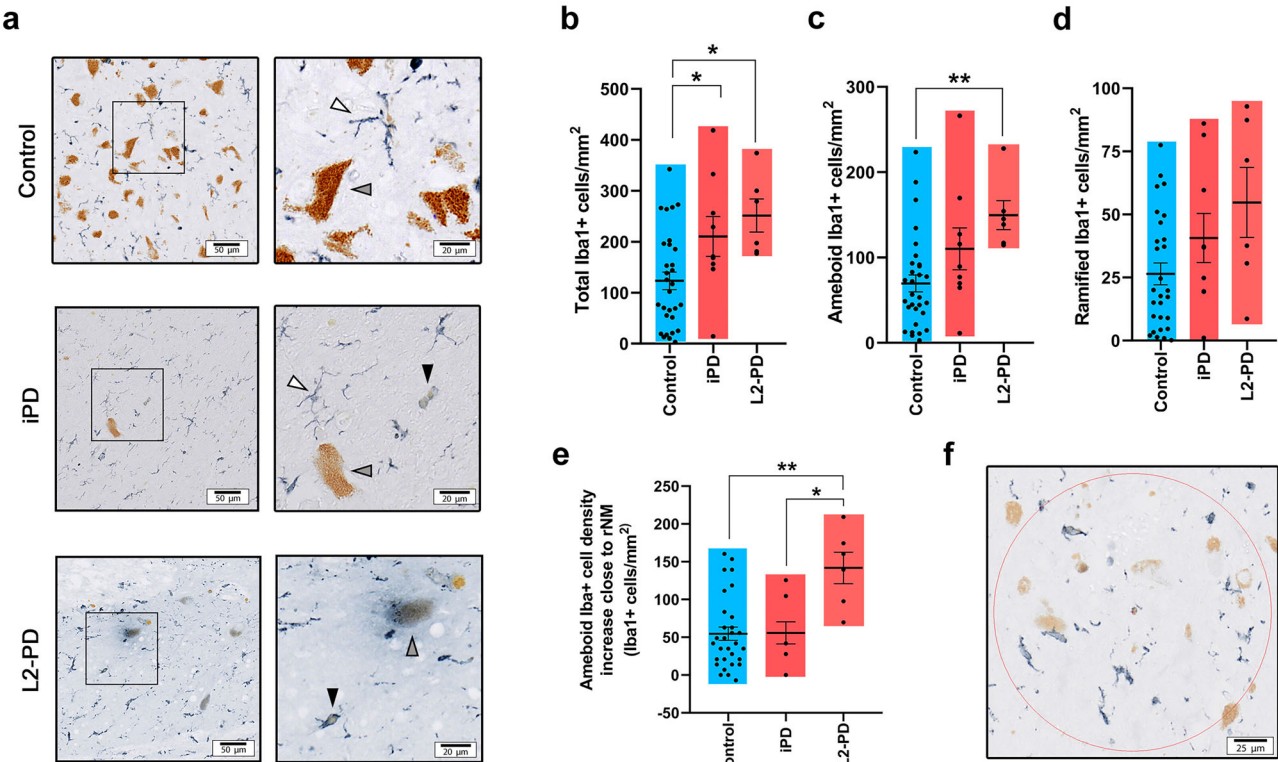

**Fig. 4 | Microglial activation is increased in L2-PD patient post-mortem brains.**
**a** Representative image of IBA-1+ cells (in blue; unstained NM in brown) from control, iPD and L2-PD human brains showing ramified (white arrowhead) or amoeboid (black arrowhead) morphologies. Grey arrowheads indicate NM-laden neurons. Scale bar = 50 μm; 20 μm. **b** Total number of IBA-1+ cells/mm² in post-mortem human brains from control subjects, iPD and L2-PD patients. **c** Number of IBA-1+ cells/mm² displaying amoeboid morphologies in control, iPD and L2-PD brains. **d** Number of IBA-1+ cells/mm² displaying ramified morphologies in

control, iPD and L2-PD brains. **e** Difference in the number of IBA-1+ cells/mm² displaying amoeboid morphologies in areas close to rNM vs areas distant from it. **f** Representative image of a quantification area close to rNM. N = 30 Control subjects, N = 9 iPD patients and N = 6 L2-PD patients. Individual data plotted, along with mean ± SEM. Individual data plotted, along with mean ± SEM. Scale bar = 25 μm. In **b** and **c**, One-Way ANOVA with Tukey's multiple comparison test; In **e**, Kruskal-Wallis with Dunn's multiple comparisons test; *p < 0.05, **p < 0.01, ***p < 0.001. p-values over 0.1 (non-significant) are not shown.

(Sigma-Aldrich) and 2 μg/mL of Fibronectin (Sigma-Aldrich) to a density of 100,000 cells/cm². From day 20 onwards, half of B27 with factors was changed every 3-4 days until vmDAn were used for experiments.

**iPSC-derived organoid generation.** Ventral midbrain organoids (vmO) were generated from iPSC using a protocol adapted from Kriks et al.[66]. Unless specified, concentration of molecules and cytokines remained as the previous vmDAn protocol. Briefly, iPSCs were grown until 90% of confluence and 10 μM of RI (Stem Macs) was added to the culture 4/6 hours before starting the protocol. Then, cells were dissociated in 1:1 accutase/DPBS, detached and centrifuged at 800x*rpm* for 5 minutes. Cell pellet was resuspended in 1 mL of fresh mTeSR™1, passed through a 40 μm cell strainer (Clear Line®) and counted. Dissociated iPSCs were plated in ultra-low attachment 6-well plastic plates (Corning®) at a density of 4 ×106 cells per well, with mTeSR™1 medium supplemented with 10 μM of RI. From then on, cells were kept under rotation on CO2 resistant orbital shaker (Leica) at 120x*rpm* and maintained under normal incubation conditions.

After 2 days, medium was replaced with SRM supplemented with LDN and SB, a dual SMAD inhibition to promote neuronal commitment[68]. On day 4, and every 3 days, half of SRM medium was replaced with fresh factors and the addition of SAG, Purmorphamine, CHIR and AA. On day 12, SRM medium was replaced by NBN2B27-vitA, which consisted of Neurobasal™ supplemented with 1x N2, 2x B27™-vitA, 1% of P/S and 1% of Glut. N2B27-vitA was changed every other day by the addition of 100 nM of LDN, 1 μM of SAG, 2 μM of Purmorphamine, CHIR and AA, 50 ng/mL of FGF-2, and 50 ng/mL of Endothelial Growth Factor (EGF; R&D Systems), to promote conversion towards a DAn fate[64]. From day 22 onwards, spheres were

matured to vmO in B27-vitA supplemented with BDNF, GDNF, TGF-β3, cAMP and AA. Half of the medium was changed every 2-3 days and vmO were cultured until day 45, before being used for experiments.

## 2D and 3D Neuron/Microglia co-culture systems
In order to visualize iPSCs-derived hMG within 2D/3D co-cultures, iPSCs were transduced with a lentiviral vector (LV) expressing Green Fluorescence Protein (GFP) under the Phosphoglycerate Kinase (PGK) ubiquitous promoter as previously described[69]. Briefly, iPSCs were detached and 10⁶ iPSCs were infected in 100 uL of mTeSR™ within a 15 mL Falcon tube with a mean of infection (MOI) of 10. Cells were incubated under normal conditions for 1 hour and plated in a well of a 6 well plate with 1 mL of mTeSR overnight (ON). After 16-18 hours, 1 mL of mTeSR™ was added. Fluorescence was checked after 72 hours under an inverted microscope.

To study neuron/microglia interactions, we generated a 2D co-culture system with iPSC-derived vmDAn. First, mature GFP-iPSC-derived hMG were conditioned with medium from vmDAn without B27-vitA, since it contains several products that could compromise microglial functioning[70]. One-week later, hMG were detached with accutase, counted and plated on top of day 42 vmDAn on a human astrocyte feeder layer, at a ratio of 1 hMG per every 2 neurons. An astrocyte feeder layer was employed for every analysis where we needed to assess neuronal health. Co-culture medium consisted of RPMI with Neurobasal™ (50/50%) supplemented with 1% Glut, 1% P/S, M-CSF, IL-34. Half of the medium was changed every 2-3 days. vmDAn and hMG were co-cultured for 1 week in hypoxia conditions, before being used for experiments. For culture treatments, 0.5 ug/mL of NM was added to co-cultures for 8 hours, and after 2 hours IVM 3 uM (Sigma Aldrich) was added, for a total of 6 hours. In case of PFF, 10 ug/mL were

added for 8 hours. Images were taken under a 63x objective employing a confocal microscopy. vmDAn morphology and neurite complexity were assessed by ICC staining of TH and analyzed by Sholl analysis 4.2.1[71] (RRID:SCR_022758, https://imagej.net/plugins/sholl-analysis) using the Simple Neurite Tracer v4.2.1 (RRID:SCR_016566; http://imagej.net/Simple_Neurite_Tracer) plugin from ImageJ 2.14.0 (RRID:SCR_003070; URL: https://imagej.net/). hMG morphology was analyzed for circularity employing ImageJ 2.14.0 (RRID:SCR_003070; URL: https://imagej.net/). Reconstruction of vmDAn and hMG were performed with IMARIS 9.7.2 (RRID:SCR_007370, http://www.bitplane.com/imaris/imaris). Surface of vmDAn was generated with a Smooth of 0.103 μm and a Background subtraction of 5 μm. Surface of hMG was generated with a Smooth of 0.103 μm and a Background subtraction of 3 μm. For the 3D co-culture system, one vmO was manually placed in a well of a U 96-wp with 50.000 GFP-hMG and incubated under normal conditions in an orbital shaker for 24 hours. Then, every organoid was collected and plated on low-adhesion 6 well plates. Co-culture medium was replaced every 2-3 days. vmO and hMG were co-cultured for 1 week in normal incubation conditions in agitation, before being used for experiments. Reconstruction of hMG was done with IMARIS 9.7.2 software (RRID:SCR_007370, http://www.bitplane.com/imaris/imaris). Volume, sphericity, and Object-Oriented Bounding Box Length C were taken as parameters for hMG morphology.

### Immunocytochemistry

Cells plated in 24-well plates were fixed before any ICC staining using 4% of Paraformaldehyde (PFA; EMS) at RT for 15 minutes. Excess of PFA was removed from the culture wells with three washes of 15 minutes with DPBS. For whole vmO IN TOTO ICC, samples were fixed in PFA at 4 °C for 2 hours and rinsed 3 times with TBS for 20 minutes. For ICC of cryosectioned vmO, samples were fixed in 4% PFA at 4 °C Overnight (ON). Fixed vmO were transferred to 30% Sucrose (Sigma-Aldrich) solution at 4 °C ON. vmO were placed on plastic molds with O.C.T.™ compound (Tissue-Tek®) at -80 °C until processing. Cryopreserved vmO were sectioned (20 μm thick) on a cryostat (Leica) and mounted in Menzel-Gläser Superfrost® PLUS coverslips (Thermo Fisher Scientific), dried at RT ON and stored at -20 °C for further ICC staining.

To stain hMG, we used a low Triton X-100 (Triton; Sigma-Aldrich) protocol. Coverslips were placed on Menzel-Gläser Superfrost® Microscope slides (Thermo Fisher Scientific) inside a humid chamber. Samples were then rinsed in 1X Tris-Buffered Saline (TBS) before being blocked at RT for 2 hours with Blocking Solution (BS), which consisted of TBS with 0.01% of Triton and 3% of Donkey Serum (DS, Millipore, Cat# S30-KC; RRID:AB_2810235). After that, samples were incubated with primary antibodies at 4 °C for 48 hours (see Supplementary Table 1). Samples were rinsed and blocked again with BS at RT for 1 hour and incubated with secondary antibodies for 2 hours at RT protected from light. All secondary antibodies employed were from Alexa Fluor Series (Jackson ImmunoResearch Europe : Cy3 Anti-Rabbit (Host: donkey; Cat. Number: 711-165-162; RRID: AB_2307443); AF488 Anti-guinea pig (Host: donkey; Cat. Number: 706-485-148; RRID: AB_2617153); AF488 Anti-mouse (Host: donkey; Cat. Number: 715-545-150; RRID: AB_2340846); DyLight 488 Anti-chicken (Host: donkey; Cat. Number: 703-486-155; RRID: AB_2736851; AF647 Anti-mouse (Host: donkey; Cat. Number: 715-605-150; RRID: AB_2340862) at a concentration of 1:250. After that, samples were rinsed with TBS and nuclei were counterstained with 0.5 μg/mL of 4',6-Diamidino-2-phenylindole (DAPI; Abcam) at RT for 10 minutes. Samples were mounted within a glass coverslip (Corning®) with Polyvinyl Alcohol-1,4-Diazabicyclo-Octane (PVA-DABCO; Sigma-Aldrich). Mounted slides were dried at RT for 30 minutes protected from light and stored at 4 °C before analysis. To stain vmDAn, 2D co-cultures or vmO cryosections, we used a high Triton protocol. In this way, the protocol timing and procedures were the same as explained, except that rinses were done with TBS with 0.1% of Triton and BS consisted of TBS with 0.3% of Triton and 3% of DS.

For the whole organoid structures, IN TOTO ICC staining was performed. vmO were rinsed with TBS and incubated with an antigen retrieval buffer composed by Sodium Citrate Tribasic (Sigma-Aldrich) with a pH of 9 at 60 °C for 1 hour. vmO were rinsed in TBS at RT in agitation (Labnet International) and permeabilized with TBS plus 0.5% Triton and 6% DS at 4 °C ON in agitation. Samples were then incubated with primary antibodies (see Supplementary Table 1), diluted in BS (TBS with 0.1% Triton and 6% DS) at RT at 4 °C over-the-weekend in agitation. After 3 days, samples were rinsed with BS at RT and incubated with secondary antibodies (diluted in BS) at 4 °C ON in agitation. The day after, vmO were rinsed in TBS and nuclei were counterstained with 2.5 μg/mL DAPI (diluted in TBS). Samples were mounted with PVA-DABCO in a glass coverslip, dried and stored at 4 °C for further imaging.

Images were acquired using Zeiss AXIOIMAGER Z1 with an Apo-Tome (Zeiss Microscopy) microscope using a Cascade CCD camera (Photometrics). For confocal images, Zeiss AXIOIMAGER Z1 with an SPE confocal system or a Zeiss LSM 880 were employed (Zeiss Microscopy). Image acquisition was realized with ZEN pro software 3.4 (Zeiss Microscopy, RRID:SCR_013672, https://www.zeiss.com/microscopy/en/products/software/zeiss-zen.html) and subsequently analyses were performed using ImageJ 2.14.0 (NIH, RRID:SCR_003070, https://imagej.net/), Photoshop® 21.0.1 (Adobe, RRID:SCR_014199, https://www.adobe.com/products/photoshop.html) and Imaris 9.7.2 (Bitplane copyright, RRID:SCR_007370, http://www.bitplane.com/imaris/imaris).

### Flow cytometry

Cells were detached, centrifuged at 300x$g$ for 10 minutes and pellet was incubated with 1:50 conjugated antibody in MACS buffer (DPBS + 0,1% BSA (Thermo Fisher Scientific)) for 15 minutes protected from the light (see Supplementary Table 2). After that, cold MACS buffer was added, and cells were centrifuged at 300x$g$ for 10 minutes. Cell pellet was resuspended in cold MACS buffer and incubated with 1:500 of Propidium iodide (Pi) at RT for 5 minutes to determine cellular viability of the culture. Unstained cells were used as negative controls. Flow cytometry measurement was done with the Gallios cytometer (Beckman Coulter Life Sciences) with the specific filters for each fluorochrome. Analyses of staining percentage and fluorescence intensities were done with Kaluza software v1.5 (RRID:SCR_016182, https://www.beckman.com/coulter-flow-cytometers/software/kaluza, Beckman Coulter Life Sciences).

### Gene expression analysis

RNA was extracted using a RNeasy® Mini kit (Qiagen) according to manufacturer's instructions and quantified with a NanoDrop one spectrophotometer (Thermo Fisher Scientific). Reverse transcription into complementary cDNA was performed with the Superscript III First-Strand Synthesis System (Invitrogen™) to a final concentration of 2 ng/μL. Gene expression was measured by real time quantitative Reverse Transcription Polymerase Chain Reaction (qRT-PCR). 8 ng of cDNA were loaded in every well of a MicroAmp™ Optical 384-well Reaction Plate (Applied Biosystems™) with 10 nM of forward and reverse primers (Life Technologies; see Supplementary Table 3) and PowerUp™ SYBR™ Green Master Mix (Applied Biosystems™). Plate was run in a Applied Biosystems 7900HT Fast Real-Time PCR System with 384-well Block Module v2.2.1 (RRID:SCR_018060, https://www.thermofisher.com/order/catalog/product/4351405#/4351408) following a standard cycling mode. Cycling values were normalized to the housekeeping gene β-Actin and the relative fold gene expression of samples was calculated using the $2^{-\Delta\Delta Ct}$ method.

### Cytokine array

The EMD MILLIPLEX® MAP Human Cytokine/Chemokine/Growth Factor Pane A – Immunology Multiplex Assay (Merck Millipore) was employed to simultaneously analyze TNF-α, IL-1β, IL-6, and IFNγ with Bead-Based Multiplex Assay using the Luminex® technology. Culture supernatants from non-stimulated and 24-hour LPS-stimulated hMG were collected and stored at -80 °C for long storage. Unless specified. Quality Controls (QC) and Human Cytokine Standard (STD) were prepared with deionized water by serial dilutions as manufacturer's instructions. Assay

Buffer was used for Background or STD 0. Briefly, wells were filled with 200 μL of Wash Buffer (WB), sealed and mixed at RT for 10 minutes on a Thermomixer Comfort plate shaker (Eppendorf™) at 500-800 rpm. Plate was firmly tap upside down on absorbent paper and 25 μL of each STD and QC with 25 μL of RPMI medium in duplicates. 25 μL of sample with 25 μL of Assay Buffer were added in duplicates. Finally, 25 μL of the mixed antibody-bead preparation was added to every well and incubated at 2-8 °C ON in a plate shaker.

The day after, well contents were gently removed by firmly attaching plate to a handheld magnet and washed with WB for 3 times. Then, 25 μL of Detection Antibodies were added into each well and incubated at RT for 1 hour on a plate shaker. After that, 25 μL of Streptavidin-Phycoerythrin were added into each well and incubated at RT for 30 minutes on a plate shaker. Well contents were gently removed with a handheld magnet and washed with WB for 3 times. Antibody-beads were resuspended by adding 150 μL of Sheath Fluid to all wells and incubated at RT for 5 minutes on a plate shaker. Finally, plate was run on a multiplex system Luminex® 200™ (Invitrogen™). Probe heights of every antibody were adjusted according to Luminex® recommended protocols employing the Diasorin xPONENT® software 4.2 (RRID:SCR_025653, https://int.diasorin.com/en/luminex-ltg/reagents-accessories/software, Luminex®). Mean Fluorescent Intensity of every analyte was analysed using a 5-parameter logistic or spline curve-fitting method for calculating analyte concentrations in samples.

### Synaptosome extraction

Mature iPSC-derived cortical neurons were obtained from neural progenitor cells (NPCs) as described[68,72]. Synaptosomes (SYNs) were then extracted from mature iPSC-derived cortical neurons using Syn-PER™ Synaptic Protein Extraction Reagent (Thermo Fisher Scientific). 1x of protease inhibitors (Roche) were added to the Syn-PER™ Reagent immediately before use. 200 μL of Syn-PER™ Reagent was added to each 6-wp well and cells were lifted from the surface using a cell scraper (Biologix). Samples were centrifuged at 1200xg for 10 minutes at 4 °C. Supernatant was transferred to a new tube and centrifuged at 15000xg for 20 minutes at 4 °C. Pellet containing SYNs was resuspended in Syn-PER™ Reagent. SYNs concentration measured using the Bradford assay solution (Bio-Rad Laboratories) and protein concentration was quantified by optical density in a PowerWave™ XS multiplate reader (Biotek®). Resulting SYNs were frozen in 5% (v/v) Dimethyl sulfoxide (DMSO, Sigma-Aldrich) at -80 °C for extended periods, until pHRodo™ Red (ThermoFisher Scientific) conjugation.

### Western blot

For protein isolation, cell pellets were resuspended in RIPA buffer (Tris HCl pH7 (450 mM), Triton 1%, NaCl (150 mM), EDTA (2 mM)) with 1X protease inhibitors (cOmplete™ Protease Inhibitor Cocktail, Sigma-Aldrich), sonicated, and centrifuged at 10,400xg for 10 minutes at 4 °C. Supernatant was collected and protein concentration was determined by Bradford assay (BioRad Laboratories). Protein was mixed with 6x LSB (Sigma-Aldrich) and 4% DTT (Sigma-Aldrich) and heated at 95 °C for 5 minutes in a Thermomixer. Samples were subjected to SDS-PAGE on 10 to 12% of polyacrylamide (BioRad Laboratories) and samples were run with a PowerPac Basic (Bio-Rad Laboratories) at 90-120 Volts for 60-90 minutes in a Mini-Protean® 3 Cell (Bio-Rad Laboratories). Gel was transferred at 350 Amperes for 120 minutes onto pre-activated polyvinylidene difluoride (PVDF) membranes in a Mini-Protean® 3 Cell (BioRad Laboratories). Correct protein transfer was checked by red Ponceau S Solution (Panreac). Membrane was rinsed 3 times with TBS with 0.1% of Tween20 (Sigma-Aldrich) and blocked at RT for 1 hour with 5% milk or 5% BSA and incubated with primary antibody at 4 °C ON (see Supplementary Table 4). The day after, membrane was rinsed and incubated with secondary Horseradish Peroxidase (HRP)-labelled antibody (Sigma-Aldrich, Cat. Number NA931, RRID: AB_772210) at RT for 1 hour. After rinsing, signal was developed with ECL™ start Western Blotting Detection Reagent (GE Healthcare Amersham™) and images were obtained using a ChemiDoc System (Bio-Rad Laboratories). Protein amount was expressed as a ratio between the band intensity of the protein of interest and the loading control protein β-Actin.

### pHrodo labeled synaptosome phagocytic assay

Appropriate amount of SYNs were thawed and centrifuged at full speed for 3-4 minutes at 4 °C. Pellet was resuspended and well-mixed in cold 100 mM of sodium bicarbonate, pH 8.5 to a concentration of 1 mg/mL. SYNs were then incubated with amine-reactive pHrodo Red Succinimidyl Ester (SE) dye (Thermo Scientific) for 60 minutes at RT in a twist shaker (Labnet International). Conjugated SYNs were rinsed with DPBS and centrifuged at full speed for 7 times[73]. 10 μg/mL of SYNs were added to hMG plated on a CellCarrier™-96 Ultra Microplate (Perkin Elmer®) and imaged using 20x objective of a Zeiss SP05 confocal microscopy, maintaining constant levels of temperature (37 °C) and CO₂ (5%). Four to five images were taken every minute for 5 hours using the phase-contrast and red fluorescence mode. 6 hours LPS pre-stimulated cells were employed as positive phagocytic controls. Phagocytic Index (PI) was quantified using ImageJ 2.14.0 (RRID:SCR_003070, https://imagej.net/), extracting background from the red fluorescent pHrodo. Total fluorescence per image was divided by the total number of cells per image[74]. PI values for every hour of the time lapse were plot and the area under the curve (AUC) was quantified to perform statistical analysis.

### Isolation of pure NM extracts from cultured TR5TY6 neuroblastoma cells

A stable inducible SH-SY5Y cell line expressing human tyrosinase (TR5TY6) under the transcriptional control of the T-Rex TM Tet-On system (Invitrogen) was provided by Dr. T. Hasegawa (Department of Neurology, Tohoku University, Sendai, Japan)[18]. Cells were confirmed negative for mycoplasma contamination by routine PCR analysis. Cells were maintained in low-glucose (1 g/L) Dulbecco's modified Eagle's (Gibco) medium with penicillin/streptomycin and the appropriate selection of antibiotics (7 μg/mL blasticidin and 300 μg/mL Zeocin, both from Life Technologies). A summary of the cell culture and isolation protocol can be seen in Supplementary Fig. 3. Briefly, TR5TY6 cells were seeded for NM isolation at $3 \times 10^6$ cells/plate onto 15-cm plates. At 24 h after seeding, cells were differentiated with 10 μM retinoic acid (RA; Sigma-Aldrich) for 3 days, followed by 80 nM 12-O-tetradecanoylphorbol-13-acetate (TPA; Sigma-Aldrich) for 3 extra days prior to hTyr induction with 2 μg/mL doxycycline (Sigma-Aldrich) for 6 days. Cells were harvested with medium using a cell scraper, centrifuged for 5 min at 300 g at 4 °C, and washed three times with phosphate buffer saline (PBS). Cell pellets were stored at −80 °C until the day of analysis.

NM isolation was performed as previously described, with some modifications[75]. Briefly, each 2 cell pellets (typically of $2 \times 10^7$ cells each) were thawed at RT and washed in 10 ml of 0.05 N phosphate buffer pH 7.2 and centrifuged at 10,000 x g for 15 min at RT. This washing and centrifugation step was repeated prior the sonication of the resulting pellets in 10 ml of 5 mg/ml SDS in 75 mM Tris pH 7.5 until total resuspension. After a 3 h incubation at 37 °C in a shaking water bath, samples were centrifuged at 10,000 x g for 30 min at RT. Pellets were then incubated for 3 h at 37 °C with 2 ml of 5 mg/ml SDS in 75 mM Tris pH 7.5 with 0.33 mg/ml proteinase K (Qiagen). At this point, samples were transferred to microtubes, pooled and subjected to a series of washing steps followed by centrifugations (10,000 x g for 30 min at RT) including: 1 ml of 0.9% NaCl (Sigma-Aldrich), 1 ml of water, 1 ml of methanol (Sigma-Aldrich), 1 ml of hexane (Sigma-Aldrich). A 25 μl aliquot was taken from the water solution to quantify NM concentration. After the last centrifugation of hexane-diluted samples, supernatants were discarded and pellets were allowed to dry in a fumehood, fast-frozen by liquid nitrogen immersion and stored at -80 °C. NM concentration of samples was quantified by measuring its absorbance at 405 nm with a NanoPhotomer NP80 (Implen). The calibration curve was prepared by serially diluting a stock solution of 1 mg/ml in 1 M NaOH synthetic melanin to cover the range of 1000 to 15 μg/ml. Pure NM extracts were sonicated in a UCI Ultrasonic cleaning bath sonicator for 15 minutes and

stored at -80 °C until use. For NM experiments, a 5ug/ml concentration in mono-cultures and an 0.5 ug/mL in co-cultures was used.

## Neuromelanin phagocytosis assay

For NM phagocytosis uptake analysis, we performed a time lapse imaging video. One-week hMG were plated in plastic wells of a 12 well plate and placed on a Zeiss AXIOIMAGER Z1 maintaining constant levels of temperature (37 °C) and CO$_2$ (5%). 5 ng/mL of NM were added to cultures and 2 images were taken from every well in duplicate every 2 minutes for 16 hours, using the phase-contrast mode. Cells were tracked and particles of uptaked NM were counted for every hour. Cells that died or moved out of the frame were excluded from the analysis. Accumulated NM was plotted during time and AUC was quantified to perform statistical analysis. After 24 hours in culture with NM, cell pellets and supernatants of non-stimulated and NM-stimulated cultures were collected and stored at -80 °C for further analyses.

## Motility analysis

Motility of hMG was assessed in 16 h-time lapse experiments and analysed using the Manual Tracker plug of ImageJ 2.14.0 (RRID:SCR_003070, https://imagej.net/). Analyses were done with Chemotaxis and Migration Tool free software version 2.0 (Ibidi®, RRID:SCR_022708, https://ibidi.com/chemotaxis-analysis/171-chemotaxis-and-migration-tool.html?gclid=CjwKCAjwu5yYBhAjEiwAKXk_eOEiTG2oTzme7pgYbZus8ZBKhu4S8YCWwlbQdQLdcOTWx4xTJ83T4xoCcVYQAvD_BwE). Velocity (µm/minute) and accumulated distance were evaluated. Accumulated distance refers to the path made by the cell from the starting point to the endpoint.

## Intracellular ROS measurement

Intracellular ROS was measured in hMG exposed to 5 ng/mL NM for 24 hours using 2′,7′-Dichlorodihydrofluorescein diacetate (DCFH$_2$-DA) probe (Sigma-Aldrich), which turns highly fluorescent upon oxidation. Ten minutes before adding the probe, hydrogen peroxide (1:25) was added to a well as a positive control of oxidation and absolute ethanol as negative control. Then, 500 µM of DCFH$_2$-DA were added for one hour at 37 °C. After that, cells were rinsed and recovered with a scrapper in PBS with 1% Triton. Samples were placed in CellCarrier™-96 Ultra Microplate and fluorescence was measured within a FLUOstar Omega microplate reader (BMG LABTECH) at 485 nm/530 nm. Results were normalized to protein concentration measured using a Micro BCA™ Protein Assay Kit (Thermo Fisher Scientific).

## Preparation of α-synuclein pre-formed fibrils (PFF)

α-synuclein pre-formed fibrils (PFF) were prepared as previously described[76]. Briefly, α-synuclein was dissolved in PBS buffer and placed in 96 wells plates containing teflon polyball (1/8″ diameter). Plates were fixed into an orbital culture shaker Max-Q 4000 Thermo Scientific (Waltham, MA, USA) to keep the incubation at 37 °C, 100 rpm for 24 hours. After production, PFF were briefly sonicated and added to the co-cultures for 8 hours at a concentration of 10ug/ml.

## Human post-mortem brain tissue

Paraffin-embedded midbrain sections (5 µm) from PD patients with LRRK2 G2019S mutation (L2-PD; n = 6) and age-matched control individuals (n = 32) were provided by the Neurological Tissue BioBank at IDIBAPS-Hospital Clinic (Barcelona, Spain). Detailed information about these subjects is provided in Table 2. Informed written consent was obtained from all human subjects. All procedures were conducted in accordance with guidelines established by the BPC (CPMP/ICH/135/95) and the Spanish regulation (223/2004) and approved by the Vall d'Hebron Research Institute (VHIR) Ethical Clinical Investigation Committee [PR(AG)370/2014]. Samples were deparaffinized at 63 °C for 30 minutes and quenched in xylene 2 times for 10 minutes at RT. To hydrate the tissue, samples were sequentially transferred to a 100% ethanol solution for 10 minutes, a 95% ethanol solution for 10 minutes and a 70% ethanol solution for 5 minutes. Samples were then rinsed for 5 minutes 3 times in 1X Tris-Buffered Saline (TBS) and

the endogenous peroxidase blocked for 15 mins using a 1X TBS solution with 3% H$_2$O$_2$ and 10% (vol/vol) methanol. Samples were rinsed again for 5 minutes 3 times in 1X TBS. Epitope unmasking for 20 minutes with 10 mM citrate buffer pH6 at 95 °C was followed by cooling for 20 minutes at RT. Samples were rinsed for 5 minutes 3 times with 1X TBS – Triton 0.5% buffer. Samples were then blocked for 1 hour using 1X TBS with 5% normal goat serum (NGS; Palex). Samples were subsequently incubated with anti-IBA1 primary antibody (host: rabbit; reactivity: human; concentration: 1:500; source: ; Abcam Cat# ab178846, RRID:AB_2636859) in 1X TBS with 2% NGS ON at 4 °C. Samples were rinsed for 5 minutes 3 times in 1X TBS before incubating with a biotinylated secondary antibody at a concentration of 1:1000 (Vector Laboratories Cat# BA-1000, RRID:AB_2313606) in 1X TBS with 2% NGS for 1 hour at RT. After rinsing for 5 minutes 3 times with 1X TBS samples were incubated with the Ultra-Sensitive ABC Peroxidase Standard Staining Kit (32050, Thermo Fisher Scientific) for 1 hour. Samples were once again rinsed for 5 minutes 3 times with 1X TBS. Vector SG Peroxidase Substrate Kit (Vector Laboratories, Cat# SK-4700; RRID:AB_2314425) was the chromogen used for 15 minutes to stain the tissue. Samples were rinsed for 5 minutes 3 times. To dehydrate the tissue, samples were sequentially transferred to a 70% ethanol solution for 2 minutes, a 95% ethanol solution for 2 minutes, a 100% ethanol solution for 2 minutes and 2 times to a xylene solution for 5 minutes. Finally, the slides were coverslipped using the DPX mounting medium (Sigma-Aldrich).

For Image acquisition and cell quantification, brain slides were scanned at 20x high resolution using an Olympus Slideview VS200 slide scanner. Scanned sections were then analysed with a specifically trained AI-powered algorithm using the Olympus V200 Desktop 3.3 software to quantify IBA-1+ cells and their morphology (i.e. ramified, amoeboid) as previously reported[19]. In order to specifically analyse IBA1+ cell counts and morphology close to and distant from rNM, eight identical circles (radius ≈ 100 µm) were drawn in each sample. Four of them were placed around randomly selected rNM granules, while the remaining four were placed in areas of the SN that were far from any rNM granule. Cells within these eight circular areas were then counted and morphologically categorized using the software mentioned above. Finally, for each brain the differences in IBA1+ cell densities close to and distant from rNM were calculated using the following formula: [(Iba1+ cells/mm$^2$ close to rNM) – (Iba1+ cells/mm$^2$ far from rNM)]. All counts were manually verified by a researcher blinded to the experimental groups.

## Statistics and reproducibility

Statistical analyses were performed using GraphPad Prism version 8.0.1 for MacOSX, (GraphPad Software, Boston, Massachusetts USA, RRID:SCR_002798; URL: www.graphpad.com). Outlier values were determined using Grubbs' test with α set to 0.05. Data normality was assessed with Shapiro-Wilk test for n < 10 or by Kolmogorov-Smirnov or D'agostino-Pearson test for n > 10. For normally distributed data, pairwise comparisons were done using two-tailed t-test. For data comprising more than two groups, we used one-way ANOVA followed by a post-hoc test, as specified in figure legends. For data departing from normality, we used the Mann-Whitney test when two groups were compared, or the Kruskal-Wallis followed by a post-hoc test as specified in figure legends when comparing more than two groups.

## Reporting summary

Further information on research design is available in the Nature Portfolio Reporting Summary linked to this article.

## Data availability

The source data behind the graphs in the paper can be found in Supplementary Data 1. Source data and key resources used in the current study are available as indicated in the Key Resource table posted at the Zenodo repository at https://doi.org/10.5281/zenodo.16601152. No code was generated for this study; all data cleaning, pre-processing, analysis, and visualization was performed using GraphPad Prism 8.0.1 (RRID:SCR_002798; http://www.graphpad.com/), Olympus V200 Desktop 3.3, Zeiss

AXIOIMAGER Z1, ZEN pro software 3.4 (Zeiss Microscopy, RRID:SCR_013672, https://www.zeiss.com/microscopy/en/products/software/zeiss-zen.html), ImageJ 2.14.0 (NIH, RRID:SCR_003070, https://imagej.net/), Photoshop 21.0.1 (RRID:SCR_014199, https://www.adobe.com/products/photoshop.html), Imaris 9.7.2 (RRID:SCR_007370, http://www.bitplane.com/imaris/imaris), Kaluza v1.5 (RRID:SCR_016182, https://www.beckman.com/coulter-flow-cytometers/software/kaluza) software, 7900HT Fast Real-Time PCR System with 384-well Block Module v2.2.1 (RRID:SCR_018060, https://www.thermofisher.com/order/catalog/product/4351405#/4351408), Luminex® 200TM, and xPONENT® (RRID:SCR_025653, https://int.diasorin.com/en/luminex-ltg/reagents-accessories/software) software. All detailed protocols have been uploaded in the public repository Protocols.io under the accession code: https://doi.org/10.17504/protocols.io.q26g7n8e9lwz/v1. Additional request of information should be addressed to the corresponding author A.C. (consiglio@ub.edu).

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

## Acknowledgements

The authors are indebted to the patients with PD who have participated in this study; the Neurological Tissue Bank of the Biobank-Hospital Clinic-IDIBAPS (Barcelona, Spain) for human sample procurement. The authors thank Giulia Carola for helping with some culture experiments. Research from the authors' laboratories is supported by the Spanish Ministry of Science and Innovation-MICINN (PID2019-108792GB-I00, PID2020-116339RB-I00, PID2021-123925OB-I00, PID2022-137963OB-I00 and PID2022-139546OB-I00 supported by MCIN/AEI/10.13039/501100011033 and FEDER, and PDC2021-121051-I00 supported by MCIN/AEI/10.13039/

501100011033 and by the European Union Next Generation EU/PRTR); Instituto de Salud Carlos III-ISCIII/FEDER (Red de Terapia Celular- TerCel RD16/0011/0024); AGAUR (2021-SGR-974), the Marató de TV3 Foundation (202012-32 and 202331-30); CERCA Program/Generalitat de Catalunya; Aligning Science Across Parkinson's through The Michael J. Fox Foundation for Parkinson's Research, USA (ASAP-020505 to M.V.); La Caixa Bank Foundation, Spain (Health Research Grant, ID 100010434 under the agreement LCF/PR/HR17/52150003 to M.V.). L.B. was the recipient of a pre-doctoral fellowship FPI (BES-2017-080579) from the Spanish Ministry of Economy and Competitiveness (MINECO). V.T. and J.C. were received a pre-doctoral La Caixa INPhINIT Incoming Fellowship (code: LCF/BQ/DI21/11860038 for V.T. and code: LCF/BQ/DI18/1166063 for J.C.). J.M.M. and V.B. were recipients of FPI pre-doctoral fellowships (PRE2022-104573 and PRE2020-094465, respectively) from the Spanish Ministry of Economy and Competitiveness (MINECO). S.V. and A.C. are recipients of an ICREA "Academia" Award (Generalitat de Catalunya). Illustrations were created in Biorender (RRID:SCR_018361, http://biorender.com: https://biorender.com/5uwr94s (Illustrations in Main Figures); https://biorender.com/4333p0o (Illustrations in Supplementary Figs. 2 and 3); https://biorender.com/fixefin (Illustrations in Supplementary Figs. 4–6); https://BioRender.com/3gtyxwg (Illustration in Graphical Abstract)).

## Author contributions

A.C., A.R., M.V. conceived the study, managed the project progress, and coordinated the experiments and analyses; L.B., M.P.-E., V.T., G.R., J.M., I.F., V.B., Y.R.-P., M.G.-S., S.J., T.C., J.M.C.-S., J.C., Z.M.-A. performed the experiments and S.V., M.J., E.T. contributed to data analyses; S.V., A.C., A.R., M.V. provided resources. The paper was prepared by L.B., M.P.E., V.T., G.R., A.R., M.V., A.C. with feedback from all authors. All authors have read and approved the current version of the manuscript.

## Competing interests

The authors declare no competing interests.

## Additional information

¹Department of Pathology and Experimental Therapeutics, Bellvitge University Hospital-IDIBELL, Hospitalet de Llobregat, Spain. ²Institute of Biomedicine of the University of Barcelona (IBUB), Barcelona, Spain. ³Neurodegenerative Diseases Research Group, Vall d'Hebron Research Institute (VHIR)-Network Center for Biomedical Research in Neurodegenerative Diseases (CIBERNED), Barcelona, Spain. ⁴Aligning Science Across Parkinson's (ASAP) Collaborative Research Network, Chevy Chase, MD, USA. ⁵Regenerative Medicine Program, Bellvitge Biomedical Research Institute (IDIBELL), and Program for Clinical Translation of Regenerative Medicine in Catalonia (P-CMRC), Hospital Duran i Reynals, Hospitalet de Llobregat, Barcelona, Spain. ⁶Centre for Networked Biomedical Research on Bioengineering, Biomaterials and Nanomedicine (CIBER-BBN), Madrid, Spain. ⁷Institut de Biotecnologia i de Biomedicina and Departament de Bioquímica i de Biologia Molecular, Universitat Autònoma de Barcelona, Bellaterra (Barcelona), Spain. ⁸Hospital Universitari Parc Taulí, Institut d'Investigació i Innovació Parc Taulí (I3PT-CERCA), Universitat Autònoma de Barcelona, Sabadell, Spain. ⁹Immunology Department–CDB, Hospital Clinic de Barcelona, Institut d'Investigacions Biomèdiques August Pi i Sunyer (IDIBAPS), University of Barcelona (UB), Barcelona, Spain. ¹⁰Department of Neurology, Hospital Clínic de Barcelona, Institut d'Investigacions Biomèdiques August Pi i Sunyer (IDIBAPS), University of Barcelona (UB), Barcelona, Spain. ¹¹Catalan Institution for Research and Advanced Studies (ICREA), Barcelona, Spain. ¹²Department of Biochemistry and Molecular Biology, Neuroscience Institute, Autonomous University of Barcelona, Barcelona, Spain. ¹³These authors contributed equally: Lucas Blasco-Agell, Meritxell Pons-Espinal, Veronica Testa, Gerard Roch. ✉e-mail: araya@idibell.cat; miquel.vila@vhir.org; consiglio@ub.edu

