## [Transparent Peer Review file · Communications Biology]

LRRK2-mutant microglia and neuromelanin synergize to drive dopaminergic neurodegeneration in an iPSC-based Parkinson's disease model

Corresponding Author: Professor Antonella Consiglio

Version 0:

Reviewer comments:

Reviewer #1

(Remarks to the Author)

The study by Blasco-Agell, Pons-Espinal, Testa, Roch and colleagues is well written with the aim of this research focused on an incompletely understood area of the field and as such this work is much needed.

The authors find that the phagocytic activity of human Lrrk2, microglia depends on G2019S mutation with the authors conclusion agreeing with the data. The Lrrk2 microglia have a proinflammatory phenotype with the authors then showing that treatment with Ivermectin rescues.

However, a significant limitation of this study concerns the DA neurons, and thus the claims about the NM treatment of microglia being significant in driving the degeneration of DA neurons. Unfortunately, considering that ~5% of the neurons generated in this study are DA neurons (Fig 3) the authors cannot make this claim. It is impossible to determine if NM treatment for the Lrrk2-PD neurons reduces the DA neurons or this finding is an artifact of very low differentiation efficiency and variability found within directed differentiation protocols. Having such a low differentiation efficiency to DA neurons is too great a limitation of this study and affects the author claims. In addition, there is no evidence that there are vmdA neurons in culture? A marker that validates the ventral midbrain (FOXA2) is lacking.

Other

- The difference between the 2 Lrrk2 patients, for instance in Figures 2C, 2E, 2G, 2I are substantial – can the authors explain this variability and comment in the discussion? Are the measurable parameters appropriate to determine phenotype?
- Please make it clear throughout the manuscript (main figures and supplementary) which cell lines the authors are referring to, L2-PD and gene-corrected, patient 1 or patient 2.
- Supp. 2e-f (inflammasome related genes vs Ctrl only, not vs isogenic). Please include this,
- Supp. 2h – Make sure the figure is properly labelled (S2h) Neurons?
- In the last section of the results – the authors refer to Supplementary Table 2 containing the information regarding the SN of the brain and controls. Supp. Table 2 contains the description of the antibodies, - Table 2 refers to the brains – please correct.
- There are two different Figure 4's included with the submission which is confusing– the figure 4 that corresponds to the manuscript does not include the figures d-f. Please amend.

Reviewer #2

(Remarks to the Author)

This research article investigates the role of neuromelanin (NM) and LRRK2-mutant microglia in the pathogenesis of Parkinson's disease (PD). Using a model derived from induced pluripotent stem cells (iPSCs), the authors examined the interaction between microglia and dopamine neurons in the presence of NM. Their findings reveal that LRRK2-mutant microglia become hyperactive upon exposure to NM, resulting in increased inflammation and neuronal death. This effect was found to be specific to NM and not associated with α -synuclein fibrils and could be reversed by ivermectin treatment. Additionally, post-mortem brain analyses corroborated the in vitro findings, showing heightened microglial activation near NM in LRRK2-PD patients.

The experiments and results presented in this article are both robust and compelling, delivered in a well-structured and coherent format. The manuscript is well-written, and the findings are communicated effectively. Based on the quality of the data and its presentation, this reviewer recommends the manuscript for publication.

Some minor comments and observations are provided below for consideration:

The decision not to use PD neurons is understandable, as the study focuses on mechanisms dependent on MGL. However, it would be interesting to explore whether the susceptibility of dopamine neurons changes under pathological conditions. It is conceivable that they might also undergo degeneration due to preformed fibrils (PFF), although it seems evident that this would not depend on MGL.

While Figure 1 clearly demonstrates that the corrected cell line responds similarly to the healthy control, it would have been helpful to maintain the comparison with the healthy control in Figure 2 as well. Nevertheless, this is not considered a significant issue.

The text references a supplementary file, "Movie S1," but this file was not provided to the reviewer.

Reviewer #3

(Remarks to the Author)

In the manuscript, Blasco-Agell et al. demonstrate that NM and LRRK2 mutations synergistically induce DA neuron degeneration using iPSC models. Overall, the study presents interesting findings. However, the synergistic effect could be better demonstrated. Detailed comments are provided below:

- Figure 2: Why were proinflammatory cytokines not detected/shown, as in Figure 1? Given their role as direct inflammation markers, their inclusion would strengthen the analysis.
- Neuromelanin can activate wild-type microglia, as shown in prior studies and Figure 2, although less prominently than LRRK2 mutant microglia. The authors should clarify why NM treatment did not induce neuronal loss in Figure 3e.
- In Figure 3i, NM-activated L2-PD hMG appears to lack neurotoxicity compared to the non-treated condition, contradicting Figure 3g. The authors should explain this discrepancy.
- Figure S4j suggests that conditioned medium mediates NM- and LRRK2 mutation-induced hMG neurotoxicity. The rescue effect of ivermectin could be better demonstrated in this model compared with in Figures 3h and 3i, given ivermectin's direct neuroprotective effects.
- To further support the synergistic effect, it would be interesting to test whether a LRRK2 inhibitor rescues NM- and LRRK2 mutation-induced hMG neurotoxicity.
- Figure 4: Using idiopathic PD postmortem samples as controls could better help assess differences of NM effects on wild-type versus LRRK2 hMG.
- The quantification of IBA-1+ cells distant from NM remains unclear despite the method description: "...The resulting density was then subtracted from the total average density to obtain the density of IBA-1+ cells distant from extracellular NM."

Version 1:

Reviewer comments:

Reviewer #2

(Remarks to the Author)

Thank you for having addressed all the points raised by me and the other reviewers. I see that now the paper is complete and I recommend for publication.

Reviewer #3

(Remarks to the Author)

The revision is acceptable, and the reviewer supports the manuscript for publication.

Point-by-point response to the reviewers' comments

REVIEWER#1

The study by Blasco-Agell, Pons-Espinal, Testa, Roch and colleagues is well written with the aim of this research focused on an incompletely understood area of the field and as such this work is much needed.

The authors find that the phagocytic activity of human Lrrk2, microglia depends on G2019S mutation with the authors conclusion agreeing with the data. The Lrrk2 microglia have a proinflammatory phenotype with the authors then showing that treatment with Ivermectin rescues.

We should like to thank the reviewer for their valuable comments and suggestions, which have helped us reshape our manuscript. We hope that the revised version of our manuscript, which incorporates new experimental evidence and text edits as per the reviewer's suggestions, will now be suitable for publication.

However, a significant limitation of this study concerns the DA neurons, and thus the claims about the NM treatment of microglia being significant in driving the degeneration of DA neurons. Unfortunately, considering that ~5% of the neurons generated in this study are DA neurons (Fig 3) the authors cannot make this claim. It is impossible to determine if NM treatment for the Lrrk2-PD neurons reduces the DA neurons or this finding is an artifact of very low differentiation efficiency and variability found within directed differentiation protocols. Having such a low differentiation efficiency to DA neurons is too great a limitation of this study and affects the author claims. In addition, there is no evidence that there are vmDA neurons in culture? A marker that validates the ventral midbrain (FOXA2) is lacking.

We thank the reviewer for this important observation and the opportunity to clarify our data on dopaminergic neuron differentiation. We would like to emphasize that our differentiation protocol is highly efficient, yielding a high proportion of vmDA neurons, with approximately 47% of MAP2+ neurons expressing TH—well in line with current standards in the field. We recognize that the way this data was initially presented may have inadvertently suggested a lower differentiation efficiency, and we are grateful for the chance to clarify this important point with additional quantifications and supporting data.

To generate the vmDA neurons, we adapted two widely used protocols (Fedele et al., *Sci Reports*, 2017 and Kriks et al., *Nature*, 2011) and have previously demonstrated their robust efficiency in our lab (Carola et al., *NPJ Parkinson's Disease*, 2021). In response to the reviewer's comment, we have now performed further characterization of the dopaminergic neuronal differentiation process. As shown in revised **Supplementary Figure 4** and detailed in the **Results (pages 6–7)**, we confirmed ventral midbrain identity by day 12 with expression of FOXA2 and LMX1A (**Figure S4b**), and by day 35, TH+/FOXA2+ neurons confirmed midbrain DA specification (**Figure S4d**). At this stage, we observed a 47% efficiency of TH+ vmDA neurons relative to MAP2+ neurons (**Figure S4e**), and a 23% efficiency relative to total DAPI+ cells (**Figure S4f**), consistent with our previous data (Carola et al., 2021).

In our co-culture system, vmDA neurons were plated on an astrocyte feeder layer and microglia were added after one week. This increases the total number of nuclei in the culture, which explains the lower TH/DAPI ratio in the final system. We agree with the reviewer that TH/DAPI ratios can be misleading in this context due to the mixed cell populations, and we apologize for the initial oversight.

To address this, we have re-analyzed all relevant data in **Figure 3** using the percentage of TH+ vmDA neurons relative to MAP2+ neurons. Under control conditions (CTRL vmDAn + ISO-PD hMG), TH+ cells make up ~41% of the neuronal population (Revised Figure 3e), closely matching differentiation yields. In the presence of L2-PD hMG, this drops from 36% (basal) to 19% with NM treatment (Revised Figure 3h), supporting our conclusion that NM-exposed L2-PD hMG are neurotoxic to vmDA neurons. For transparency, we have also

included the original TH/DAPI quantifications in revised **Supplementary Figure 4** (panels g–j).

Other

- *The difference between the 2 Lrrk2 patients, for instance in Figures 2C, 2E, 2G, 2I are substantial – can the authors explain this variability and comment in the discussion? Are the measurable parameters appropriate to determine phenotype?*

We thank the reviewer for the thoughtful comment and fully agree with the concern. In the revised manuscript (**Discussion, page 9**), we now directly address the issue of donor-specific variability, a known limitation in iPSC-based models (Kilpinen et al., *Nature* 2017; Guhr et al., *Stem Cell Reports* 2018). To address this, we included isogenic control lines in our study, allowing us to assess the specific impact of the LRRK2-G2019S mutation while minimizing variability from the genetic background.

We also appreciate the reviewer's suggestion regarding the selection of appropriate parameters to assess phenotype. Upon re-evaluation, we recognized that one of the analyses—specifically the velocity readout shown in the original Figure 2e—could give the misleading impression that the two L2-PD lines behaved differently. To avoid confusion, we have removed this panel from the revised version.

Importantly, as shown in revised Figures 2f, 2g, 2i, and 2k, both L2-PD lines show a consistent shift toward a pro-inflammatory profile upon NM exposure when compared to their respective isogenic controls. While the degree of response varies, the overall trend is shared: L2-PD1 shows a stronger induction of TNF- α and IL-1 β , whereas L2-PD2 responds more markedly with increased C3, IL-6, and IFN γ . Together, these results support a common inflammatory phenotype associated with the LRRK2 mutation, while also reflecting individual variability. We hope these changes clarify our approach and strengthen the conclusions of the manuscript.

- *Please make it clear throughout the manuscript (main figures and supplementary) which cell lines the authors are referring to, L2-PD and gene-corrected, patient 1 or patient 2.*

We have revised all figure legends including main and Supplementary figures and included the name of the iPSC lines used for each experiment in the revised version of the manuscript.

- *Supp. 2e-f (inflammasome related genes vs Ctrl only, not vs isogenic). Please include this.*

We thank the reviewer for the suggestion and we performed a quantitative real time PCR to measure the inflammasome related gene expression in CTL (SP09), L2-PD1 (SP12) and L2-PD1^{corr} (SP12wt/wt) hMG. Results are included in new **Supplementary Figure 2e-f**.

- *Supp. 2h – Make sure the figure is properly labelled (S2h) Neurons?*

We appreciate the reviewer's point, which we have corrected in the revised version of our manuscript in new **Supplementary Figure 2h**.

- *In the last section of the results – the authors refer to Supplementary Table 2 containing the information regarding the SN of the brain and controls. Supp. Table 2 contains the description of the antibodies, - Table 2 refers to the brains – please correct.*

We thank the reviewer for noticing these mistakes, which have been corrected in the revised version of our manuscript (**page 8** of the revised manuscript).

- *There are two different Figure 4's included with the submission which is confusing– the figure 4 that corresponds to the manuscript does not include the figures d-f. Please amend.*

We apologize for the mistakes, which have been corrected in the revised version of our manuscript.

This research article investigates the role of neuromelanin (NM) and LRRK2-mutant microglia in the pathogenesis of Parkinson's disease (PD). Using a model derived from induced pluripotent stem cells (iPSCs), the authors examined the interaction between microglia and dopamine neurons in the presence of NM. Their findings reveal that LRRK2-mutant microglia become hyperactive upon exposure to NM, resulting in increased inflammation and neuronal death. This effect was found to be specific to NM and not associated with α -synuclein fibrils and could be reversed by ivermectin treatment. Additionally, post-mortem brain analyses corroborated the in vitro findings, showing heightened microglial activation near NM in LRRK2-PD patients.

The experiments and results presented in this article are both robust and compelling, delivered in a well-structured and coherent format. The manuscript is well-written, and the findings are communicated effectively. Based on the quality of the data and its presentation, this reviewer recommends the manuscript for publication.

We are very glad to read that the reviewer appreciated the interest of our studies. We should like to thank the reviewer for his/her comments and suggestions, and hope that the revised version of our manuscript, which incorporates new experimental evidence and text edits as per reviewer's recommendations, will satisfactorily address his/her concerns.

Some minor comments and observations are provided below for consideration:

The decision not to use PD neurons is understandable, as the study focuses on mechanisms dependent on MGL. However, it would be interesting to explore whether the susceptibility of dopamine neurons changes under pathological conditions. It is conceivable that they might also undergo degeneration due to preformed fibrils (PFF), although it seems evident that this would not depend on MGL.

We appreciate the reviewer's suggestion, which we have addressed in the revised version of our manuscript (**Discussion section: page 11**).

As the reviewer suggested, it would be interesting to evaluate whether iPSC-derived dopaminergic neurons from L2-PD patients exhibit the same sensitivity to PD-related pathological insults or display an exacerbated phenotype upon stimulation with preformed fibrils or NM. However, the primary focus of this study was to assess the effect of hMG on dopaminergic degeneration. Indeed, in previous studies, we showed that neurons derived from patient iPSCs recapitulate PD-relevant disease-associated phenotypes, including morphological alterations such as a reduced number of neurites, decreased neurite arborization, and the accumulation of autophagic vacuoles—features not observed in dopaminergic neurons (DAn) differentiated from control iPSCs. Therefore, in this study, to avoid cell-autonomous effects and focus exclusively on the impact of LRRK2-mutant microglia on dopaminergic neurons, we used iPSC-derived neurons from healthy individuals and exposed them to either mutant or isogenic control hMG.

While Figure 1 clearly demonstrates that the corrected cell line responds similarly to the healthy control, it would have been helpful to maintain the comparison with the healthy control in Figure 2 as well. Nevertheless, this is not considered a significant issue.

We agree with the reviewer that it would be helpful to add a comparison with the healthy control line in Figure 2. Nevertheless, due to the limited availability of MG samples treated with NM, we were unable to repeat all the experiments in Figure 2 using the CTL line. However, we were able to measure IL-6 (as requested by reviewer #3) and ROS levels for the CTL line. Results are shown in **Figure 2j and 2k** of the revised version of the manuscript.

The text references a supplementary file, "Movie S1," but this file was not provided to the reviewer.

We apologize for the mistake. We have corrected it and included a **Supplementary Movie 1** in the revised version of our manuscript.

REVIEWER#3

In the manuscript, Blasco-Agell et al. demonstrate that NM and LRRK2 mutations synergistically induce DA neuron degeneration using iPSC models. Overall, the study presents interesting findings.

We are very pleased to learn that the reviewer found our study of interest. We would like to sincerely thank the reviewer for their thoughtful comments and valuable suggestions. We have carefully addressed the concerns raised and incorporated both new experimental data and textual revisions in accordance with the recommendations. We hope that these changes will satisfactorily resolve the reviewer's comments and further strengthen the manuscript.

However, the synergistic effect could be better demonstrated. Detailed comments are provided below:

• Figure 2: Why were proinflammatory cytokines not detected/shown, as in Figure 1? Given their role as direct inflammation markers, their inclusion would strengthen the analysis.

We sincerely thank the reviewer for this valuable experimental suggestion. In response, we assessed the levels of secreted pro-inflammatory cytokines following NM stimulation in our hMG cultures. Among the cytokines measured, IL-6 was reliably detected in the culture media, and we have now included these data in the revised manuscript (**Figure 2j**). However, the other cytokines remained below the detection threshold under these conditions. This likely reflects the nature of the NM-induced inflammatory response, which appears to trigger a milder cytokine release compared to classical pro-inflammatory stimuli such as LPS. For example, as shown in **Figure 1e**, LPS stimulation results in IL-6 levels of approximately 6000 pg/mL, whereas NM treatment leads to significantly lower IL-6 secretion (~20 pg/mL, revised **Figure 2j**). Since other cytokines are typically expressed at even lower levels than IL-6 during robust LPS-induced inflammation (**Figure 1d, 1f, and Supplementary Figure 2d**), it is possible that under NM stimulation, their protein levels fall below detection limits and may only be measurable at the mRNA level, where sensitivity is considerably higher.

• Neuromelanin can activate wild-type microglia, as shown in prior studies and Figure 2, although less prominently than LRRK2 mutant microglia. The authors should clarify why NM treatment did not induce neuronal loss in Figure 3e.

The reviewer raises an interesting point, as well as a promising potential experimental direction. While this study focuses primarily on the deleterious role of mutated microglia in promoting dopaminergic neuronal loss, it is important to acknowledge that, under normal physiological conditions, activated microglial cells contribute to debris clearance while maintaining an anti-inflammatory profile. It is only when their phagocytic capacity becomes overwhelmed that excessive ROS and pro-inflammatory cytokine production begins to occur. Both microglial phenotypes can coexist, as evidenced by the moderate but significant activation observed in wild-type lines, as the reviewer rightly points out. In this context, cytokine and ROS levels may remain below the threshold required to induce neurodegeneration. However, mutations in genes such as LRRK2, which promote microglial overactivation, can severely disrupt this finely tuned balance. It is entirely plausible that, over time—and given the inability to process all the NM present in the culture—wild-type microglia might also reach the limits of their clearing capacity, adopt a pro-inflammatory profile similar to that observed in LRRK2-mutant lines, and contribute to neurodegeneration. As such, evaluating both anti-inflammatory and reparative, as well as pro-inflammatory, markers in NM-exposed microglial lines represents an experimental avenue worth pursuing.

• In Figure 3i, NM-activated L2-PD hMG appears to lack neurotoxicity compared to the non-treated condition, contradicting Figure 3g. The authors should explain this discrepancy.

We thank the reviewer for highlighting this important point. We acknowledge that the apparent discrepancy between **Figures 3g and 3i** in the previous version of the manuscript was due to

inconsistent data representation. Specifically, Figure 3i displayed data as fold changes, whereas **Figure 3g** presented results as percentages of TH⁺/DAPI⁺ cells, which may have led to confusion.

To address this concern and ensure consistency throughout, we have re-analyzed and re-plotted all relevant data using a unified metric—percentage of TH⁺/MAP2⁺ cells—as also requested by Reviewer #1. Additionally, we have merged the previous **Figures 3g and 3i** into a single, updated panel (now **Figure 3h**) to provide a more transparent and direct comparison. These changes clarify that NM-treated L2-PD hMG is indeed neurotoxic compared to the untreated condition (19% vs. 37% surviving dopaminergic neurons), thereby aligning with the findings across the dataset and resolving the previously perceived inconsistency.

• Figure S4j suggests that conditioned medium mediates NM- and LRRK2 mutation-induced hMG neurotoxicity. The rescue effect of ivermectin could be better demonstrated in this model compared with in Figures 3h and 3i, given ivermectin's direct neuroprotective effects.

The reviewer raises an important point. Ivermectin has been shown not only to restore the inflammatory profile of microglia (Zabala, EMBO Mol Med, 2018) but also to exert neuroprotective effects (Seyyedabadi, Metabolic Brain Disease, 2023; Wadsworth, Cell Biosci, 2024). Our study suggests that DA neurodegeneration is driven by pro-inflammatory L2-PD hMG exposed to NM, and we propose that ivermectin's effect is mediated by hMG. However, as the reviewer pointed out, we cannot rule out the possibility that Ivermectin also acts directly on neurons.

Unfortunately, we did not have access to the necessary materials for doing the experiments, and generating additional iPSC-derived hMG and DA neurons would have required more time than feasible within the expected timeline for this publication. Nonetheless, we sincerely appreciate the reviewer's suggestion for further validation of this hypothesis. We have addressed this point in the **Discussion section** on **page 10**).

• To further support the synergistic effect, it would be interesting to test whether a LRRK2 inhibitor rescues NM- and LRRK2 mutation-induced hMG neurotoxicity.

We appreciate the reviewer's experimental suggestion. In our study, we performed all experiments using cultures differentiated from both L2-PD iPSCs and their gene-corrected isogenic iPSC counterparts in parallel. We strongly believe that our results provide definitive evidence for the role of mutated LRRK2 in the observed phenotypes, using genetic rescue experiments that arguably go beyond the requested pharmacological interventions. Nonetheless, we have previously attempted pharmacological inhibition of LRRK2 using the compound LRRK2-IN1 (Calbiochem, cat. no. 438193) in our co-culture model of CTL neurons with L2-PD or isogenic microglia. The inhibitor was applied at concentrations of 0.05 μ M, 0.5 μ M, and 5 μ M daily for 7 days, starting at co-culture day 21. Unfortunately, under all tested conditions, including the lowest concentration and even in control co-cultures, LRRK2-IN1 treatment resulted in poor dopaminergic neuron survival, rendering interpretation of any potential rescue effects unreliable.

While we are unable to provide pharmacological rescue data at this time, we acknowledge the importance of this direction and are actively exploring alternative LRRK2 inhibitors and dosing regimens that might be more compatible with long-term neuronal health in our co-culture system.

• Figure 4: Using idiopathic PD postmortem samples as controls could better help assess differences of NM effects on wild-type versus LRRK2 hMG.

We thank the reviewer for this valuable suggestion. In response, we have included idiopathic PD (iPD) post-mortem samples as additional controls to better assess the differential effects of neuromelanin (NM) exposure in wild-type versus LRRK2 hMG.

The revised text in the **Results section** (**pag.8**) now reads:

“To determine the relevance of our in vitro findings using iPSC-derived hMG to human PD pathology, we next examined microglial activation in the SN of post-mortem brains from iPD and L2-PD patients, comparing them to age-matched controls (Table 2). BA1+ cell density was significantly elevated in both iPD and L2-PD brains compared to controls (Fig. 4a-b). More importantly, a significantly greater proportion of these microglial cells exhibited an amoeboid morphology—indicative of activation—in L2-PD cases compared to non-PD controls (Fig. 4c). Interestingly, the number of amoeboid cells in iPD patients was intermediate between that of the control and L2-PD groups. In contrast, the number of non-reactive (ramified) microglial cells did not differ among the groups (Fig. 4d).

We then sought to determine whether rNM proximity influences microglial activation differently across groups. To that end, areas deemed distant from and in close proximity to eNM were analyzed and compared (Fig. 4f). As expected, amoeboid IBA1+ cell density showed an increasing trend around rNM in all three groups. However, this effect was significantly more pronounced in L2-PD compared to both iPD and control groups (Fig. 4e). These results support our in vitro findings in NM-exposed hMG and further reinforce the idea that NM-linked microglial activation may contribute to PD pathogenesis”.

• The quantification of IBA-1+ cells distant from NM remains unclear despite the method description: “...The resulting density was then subtracted from the total average density to obtain the density of IBA-1+ cells distant from extracellular NM.”

We thank the reviewer for this insightful comment. The concern is fully justified. Initially, we estimated the density of IBA-1⁺ cells distant from extracellular NM by subtracting the local density near NM from the total average. Upon re-evaluation, we recognized the limitations of this approach and have since revised our quantification method to directly assess distant IBA-1⁺ cell density more accurately. Corresponding changes have been made to both **Figure 4** and the **Methods section** to reflect this improved analysis.

Resubmission of COMMSBIO-24-8735-T

REVIEWERS' COMMENTS:

Reviewer #2 (Remarks to the Author):

Thank you for having addressed all the points raised by me and the other reviewers. I see that now the paper is complete and I recommend for publication.

Reviewer #3 (Remarks to the Author):

The revision is acceptable, and the reviewer supports the manuscript for publication.

We appreciate the reviewers' kind comments regarding our manuscript.